# Identifying and Mitigating Model Failures through Few-shot CLIP-aided Diffusion Generation

## Abstract

Deep learning models encounter unexpected failures, especially when dealing with challenging sub-populations. One common reason for these failures is features that the data may be spuriously correlated with. To better understand these failure modes, human-interpretable descriptions are crucial, which is expensive. In this study, we propose an end-to-end framework that utilizes the capabilities of large language models (ChatGPT) and vision-language deep models (CLIP) to generate text descriptions of failure modes associated with spurious correlations (e.g., rarely seen backgrounds) without human-in-the-loop intervention. These descriptions can be used to generate synthetic data using generative models, such as diffusion models. The model can now use this generated data to learn from its weaknesses and enhance its performance. Our approach serves as a broad solution, promising progress in comprehending model failure modes and strengthening deep learning models automatically across a wide range of failure scenarios (e.g., backgrounds, colors) in a few-shot manner. Our experiments have shown remarkable **improvements in accuracy ($\sim$ 21%)** on hard sub-populations (particularly for wrong background association) across 30 different models, such as ResNets, EfficientNets, DenseNets, Vision Transformer (ViT), SwAVs, MoCos and DINOs on various datasets such as ImageNet-1000, and CIFAR-100, iNaturalist-2018.

## 1 Introduction

The quality of training data directly impacts the performance and robustness of machine learning models. Despite careful curation of training data, models can still exhibit failure modes where their performance deteriorates in specific sub-populations of data, leading to misclassifications or inaccurate predictions Jiang et al. (2018); Arpit et al. (2017). The failure modes of deep networks can arise from various factors, such as noisy labels Sukhbaatar et al. (2014); Jiang et al. (2018); Reed et al. (2015), multi-labels Zhang et al. (2018b), and spurious correlations Zhou et al. (2020), particularly when it comes to distinguishing between objects and their backgrounds Kattakinda & Feizi (2021); Xiao et al. (2021). (See Figure 7 in appendix for examples of these failures.)

Similar to how humans use image backgrounds as cues for object recognition, studies have shown that machine learning models also rely on backgrounds when making decisions. In some cases, models may prioritize backgrounds to the point of overlooking important object features for classification Zhang et al. (2007); Ribeiro et al. (2016); Sagawa et al. (2020).

Various strategies have been attempted to mitigate failure modes caused by spurious background associations, but many are insufficient in addressing the entirety of the problem. Some methods involve human-in-the-loop interventions Mitchell et al. (2021); Santurkar et al. (2021), which are both labor-intensive and challenging to scale for large operations. Furthermore, many of these approaches specifically target only one spurious correlation, such as background, and may not be readily applicable to other correlations like colors, thus neglecting a comprehensive spectrum of potential failures Barbu et al. (2019); Hendrycks et al. (2021a); Hendrycks & Dietterich (2019); Hendrycks et al. (2021b); Kattakinda & Feizi (2021). Additionally, certain existing works lack clear descriptions of model failures in a human-understandable manner, posing challenges in terms of interpretability and validation.

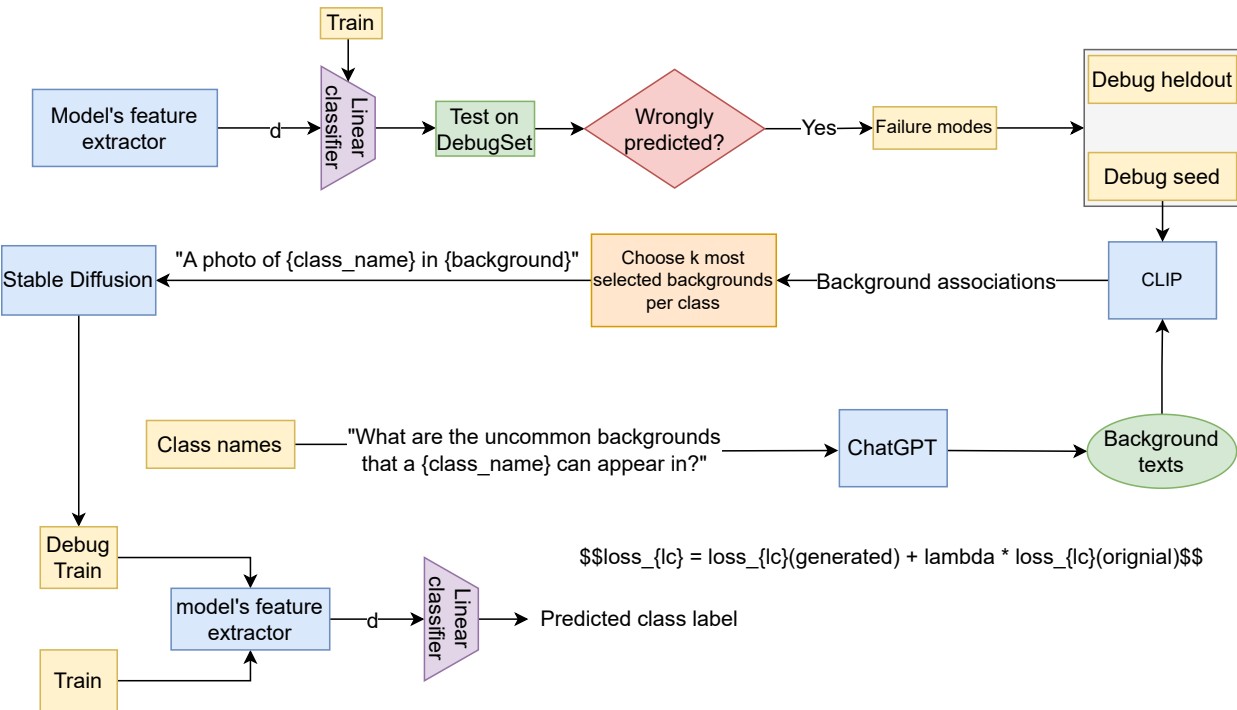

Figure 1: A summary of our approach applied to background spurious correlations: For a model based on the wrongly predicted debug samples (failure samples), we form two sets - **_DebugSeed_** and **_DebugHeldout_**. We use the **_DebugSeed_** set to address the model's failures by inputting them to CLIP Radford et al. (2021), along with a set of backgrounds obtained from ChatGPT where objects are less likely to occur with the data. We then obtain a set of backgrounds and remove redundancies, and generate synthetic data by inputting the prompt "A photo of {class_name} {background}" to Stable Diffusion Rombach et al. (2021). With this synthetic data that precisely captures the model's failure modes related to backgrounds, we can now refine the model's predictions on other test data by training a very low-cost linear head on top of our model, which assigns different weights to the original data and the generated data.

In conjunction with research on identifying failure modes, there are various **refinement** approaches aimed at leveraging these failure modes to improve the accuracy of machine learning models. These strategies involve actions such as generating additional datasets containing failure samples to assist the model in learning robust features Xiao et al. (2021); Singla et al. (2024) or adjusting the model's parameters to integrate information derived from identified failure modes Rame et al. (2022). However, these studies often lack easily understandable descriptions of failure modes for human interpretation, posing challenges in assessing their validity. Furthermore, these refinement approaches are typically not automated.

## 2   Our contribution

This research leverages recent generative models, large language models, and CLIP to introduce an automated framework addressing failure modes (spurious correlations) in diverse task-specific deep learning models. The framework, outlined in Figure 1, answers critical questions such as identifying and rectifying spurious associations leading to model failure, utilizing these failure modes to refine models, exploring patterns in failure modes across a model group, and using a single set of auxiliary data to improve a subgroup of models simultaneously.

To summarize, our approach initially identifies all model failures on a specific subset, denoted as **_DebugSet_**, which is a part of the validation set. We then pinpoint spurious correlations, such as background asso-

ciations, for each dataset class by querying ChatGPT with "What are the uncommon backgrounds that a class_name can appear in?" and remove redundancies after obtaining uncommon backgrounds for all classes. Subsequently, a zero-shot classification using CLIP identifies the background for each failure among all the uncommon backgrounds. To enhance model performance, we generate $k$ artificial images with prompts like "[class_name] in [background_name]" and incorporate this supplementary data into the original_train_set. In the second phase, we demonstrate that models with similar architectures exhibit analogous failures, allowing efficient troubleshooting of a group of models using a single set of generated auxiliary data. This approach proves both time and memory-efficient. The results of our experiments, detailed in section 5, underscore the effectiveness of this straightforward method in achieving interpretability and refinement goals.

Our paper presents several contributions to model failure analysis and refinement. These contributions include, but are not limited to, the following:

- **Generalizability**: Introducing an automatic end-to-end framework that interprets and rectifies failures arising from specific spurious associations, such as incorrect background, color, and size correlations, which can contribute to any model inaccuracies.

- **Failure Inspection**: Identification of spurious associations(section 5.2.1), and exploring common patterns in failure modes among individual models with same architectures (section 5.2.2) in an interpretable manner .

- **Failure Mitigation**: Improving the performance of individual models on challenging sub-populations (5.3.1), and boosting the performance of a subset of models by employing a unified set of auxiliary data, leveraging shared failures to enhance efficiency in both time and memory usage (section 5.3.2).

- **Collective Failure Mitigation**: Refinement of a subset of models' performance through a unified set of auxiliary data owing to their shared failures, which saves time and memory. **To the best of our knowledge, this work represents the first effort to collectively address failures within a subgroup of models simultaneously. (section 5.3.2).**

## 3 Related work

### 3.1 Failure mode detection

Numerous studies have been conducted to detect failure modes in machine learning models. Some involve human-in-the-loop methods, where failure examples are reviewed to identify common patterns Mitchell et al. (2021); Santurkar et al. (2021). Others adopt automated approaches by introducing frameworks that effectively capture model failures Chung et al. (2019); Singla et al. (2021); Nushi et al. (2018); Singla & Feizi (2022); Wong et al. (2021); Wu et al. (2019); Zhang et al. (2018a); Jain et al. (2023). For instance, Chung et al. (2019) employs a technique that slices the validation data to isolate sections where the model performs poorly. Singla et al. (2021) identifies visual attributes that lead to inadequate performance when present or absent. Jain et al. (2023) identifies and represents model failures as directions in the latent space, and Eyuboglu et al. (2022) proposes an evaluation framework to systematically compare (slice discovery method) SDMs across diverse slice settings by generating captions for hard sub-populations. Distinguishing itself from existing methodologies, our approach provides enhanced generality by **permitting the explicit selection of the spurious correlation targeted for mitigation, targeted data collection, giving interpretable descriptions for failures, and being an automatic approach**. For instance, although the approach presented by Kattakinda et al. Kattakinda et al. (2022) effectively tackles spurious correlations tied to foreground and background features by learning disentangled representations, it encounters difficulties when confronted with a broader spectrum of spurious correlations, e.g., color. This is due to the inherent challenge of learning disentangled representations for many spurious correlations in isolation from the foreground object.

### 3.2 Mitigation of Hard Subpopulations and Interpretability of Models

Several methodologies leverage extracted failure modes to enhance the performance of deep learning models. Singla et al. (2024) introduce a framework that identifies visually similar images to model failures and incorporates them as new data for refinement of various models. Kattakinda et al. (2022) focus on learning invariant features for foreground and background by penalizing the mutual information between the features and background/foreground labels. This approach contributes to robust model training, particularly by addressing issues related to spurious correlations.

In data generation, Bansal & Grover (2023), and Wiles et al. (2022) use generated data to diversify training datasets. However, it's essential to note that their methods do not specifically target failure modes like spurious correlations. They rely on class names and general captions for generating auxiliary data, which may not be tailored to address specific failure modes.

Moreover, Wiles et al. (2022) propose a bug discovery approach using off-the-shelf image generation and captioning, contributing to the interpretability of model predictions. On the other hand, Jain et al. (2023) leverage Support Vector Machines (SVMs) to distill model failures as directions in latent space, focusing on latent representations of model failures.

Compared to existing methodologies that address failure modes on specific datasets, our framework introduces two noteworthy contributions. Firstly, **it achieves enhanced model performance with significantly fewer generated examples (5 for each failure)**. Secondly, **our experiments extend to collective refinement, demonstrating the ability to improve a subset of model failures by generating a single auxiliary artificial dataset based on only one model's failures**. This is particularly valuable given our observation that models within the same categories exhibit similar failures, a phenomenon also noted in Wiles et al. (2022).

Moreover, our approach is efficient, eliminating the necessity for complete model retraining or fine-tuning. We exclusively focus on retraining the linear head for classification, streamlining the failure mode mitigation process.

### 3.3 Synthetic data as data augmentation

Numerous studies leverage the generative capabilities of diffusion and GAN models to produce synthetic data, enhancing training datasets for better accuracy in downstream tasks. For example, in work by Hong et al. (2023), a classifier is trained using consistency rules on unlabeled data generated from unconditional GANs, improving image classification. Zhou et al. (2023) employ the Stable Diffusion model to generate diverse and high-quality training data for image classification efficiently. The theoretical aspect of using synthetic data and its stability bound is explored by Zheng et al. (2023), offering insights into improved learning rates achievable with generative data augmentation, especially in small training set scenarios. Ye-Bin et al. (2023) address the data imbalance problem by generating synthetic data. Additionally, Luzi et al. (2022) introduces varied, nonidentical images through a partial reverse diffusion process, serving as a data augmentation method to enrich training datasets. Various image editing methods Meng et al. (2021); Kawar et al. (2023); Zhang et al. (2023); Brooks et al. (2023); Mokady et al. (2023); Koohpayegani et al. (2023) can also be considered for synthetic data generation to enhance training datasets. Our method uses the targeted generation of synthetic data for hard subpopulations as data augmentation to address the problem of spurious correlations.

## 4 Main method

### 4.1 Failure-mode detection

A common reason for accuracy drops during inference is the model's learned spurious correlations from training. For example, Associating objects with backgrounds, a spurious correlation can hinder the model's ability to learn about objects themselves. This challenge arises when the model encounters objects in unfamiliar backgrounds during testing, notably in computer vision tasks where backgrounds define object

context. To tackle this, introducing the model to a range of scenarios that address the particular failure mode (such as color or background associations) we aim to mitigate can improve its ability to identify objects in different contexts and avoid correlating the objects and their changeable features (e.g., color) or contexts (e.g., background).

Initially designed to rectify wrong background associations, our framework can be extended to address various spurious correlations. We showcase its applicability by presenting results for color spurious associations in 5.

To address and rectify failures attributed to backgrounds, we utilize the feature extractor for each model on the datasets, generating a feature vector for each data point. The subsequent linear head atop this feature extractor executes the classification task. Instances where the model makes incorrect predictions form a set termed the **DebugSet**, which serves as a tool for identifying and resolving failure modes, comprising all examples where the model fails. While these failures may stem from various factors, our experiments underscore the significance of mitigating incorrect background associations, as they significantly improve the performance of all models.

### 4.2 Failure-mode textualization

Vision-language models are popular as they can provide a more comprehensive understanding of complex phenomena by combining information from different modalities like text, images, and audio, enabling them to interpret data in a more human-readable form Lu et al. (2019); Chen et al. (2018); Mithun et al. (2020).

Understanding failure modes is critical for validating proposed refinement methods. By identifying the causes of failure, we can improve our models and refine our data collection methods. For each class_name in our dataset, we first prompt ChatGPT, "What are the uncommon backgrounds that a class_name can appear in?" filter out the redundant suggested backgrounds and keep the 10 suggested uncommon backgrounds for each class. Some examples can be seen in Table 1. Then, we use CLIP Radford et al. (2021) to interpret failure modes by splitting the failures from the **DebugSet** into two sets called **DebugSeed** and **DebugHeldout**. We then perform zero-shot classification by passing **DebugSeed** along the set of uncommon backgrounds proposed by ChatGPT to a CLIP model, so for each data point, CLIP will opt for the background that is more likely to be the actual background of the object shown in the image. For each data class, we then pinpoint the $k$ most frequently selected backgrounds by CLIP, which the model failed to classify. Our experiments, as depicted in Figure 5, indicate that the optimal value for k, which is both small and practical, is 3. While increasing the value of $k$ may enhance final results, the marginal improvement is negligible compared to the cost and time associated with generating additional synthetic data. This will provide valuable insights into how a model may fail when confronted with a particular selected spurious association.

| Class name | Uncommon backgrounds |
|---|---|
| Sea lion | Desert, Rain forests, Urban Areas, Polar Ice Caps, Caves, Grasslands, Volcanic Areas |
| Siberian Husky | Jungle Canopies, In the Sky, Caves, Underwater, Indoor Spaces, Marshlands, Tropical Rainforests |
| croquet ball | Mountain Peaks, Busy Streets, Frozen Lakes, Underneath Building Foundations, Subway Tunnels, in a restaurant |
| lipstick, lip rouge | Gyms and Fitness Centers, Swimming Pools, Medical Facilities, Construction Sites, Sports Events, Military Training |

Table 1: Examples of suggested uncommon backgrounds for a class of data by ChatGPT

### 4.3 Generating synthetic data

By leveraging CLIP's detected backgrounds of failures, we can interpret them and use them to refine models. For instance, in the case of the ImageNet class "tench," errors predominantly occur when the fish is held by a person's hand, a scenario rarely encountered during training. To address this, a generative model like Stable

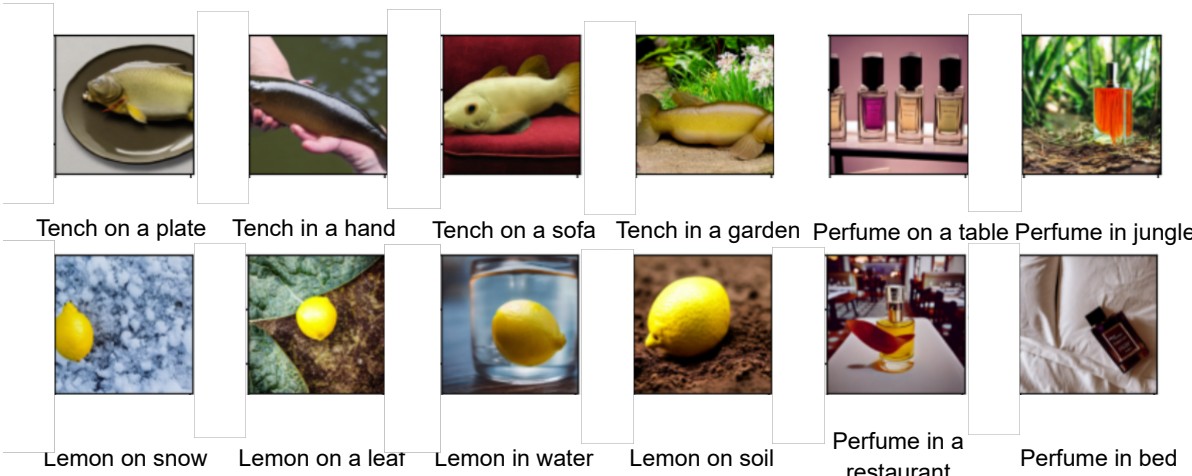

Tench on a plate   Tench in a hand   Tench on a sofa   Tench in a garden   Perfume on a table   Perfume in jungle

Lemon on snow   Lemon on a leaf   Lemon in water   Lemon on soil   Perfume in a restaurant   Perfume in bed

Figure 2: Examples of generated data by Stable Diffusion

Diffusion Rombach et al. (2021) can create images that familiarize the model with diverse object contexts. We generate data for the "tench" class by inputting the prompt "tench in hand" to the Stable Diffusion. Examples of such generated data are presented in Figure 2

### 4.4   Retraining the linear head

After collecting the additional synthetic data for the failed scenarios, which we call **DebugTrain**, we can use it along with our original_train_set to refine our models. To achieve this, we only need to train a linear head on top of the model's feature extractor for classification purposes, not the whole model. We must note that we assign different weights to the data points from the original_train_set and the **DebugTrain** set in our linear head training loss 1. This parameter is called **lambda**, and in our experiments shown in 5, we report its effect on the overall performance of the model. By incorporating the additional **DebugTrain** data and carefully tuning the **lambda** parameter, we can potentially improve the performance of our models.

$$L_{cl} = L_{cl}(Original\_train\_set) + \lambda * L_{cl}(DebugTrain) \tag{1}$$

## 5   Experiments

### 5.1   Setting

We conducted experiments on 40 pretrained models, including ResNets He et al. (2016), EfficientNets Tan & Le (2019), DenseNets Huang et al. (2017), Vision Transformer (ViT) Dosovitskiy et al. (2021), SwAVs Caron et al. (2020), MoCos He et al. (2019), DINOs Caron et al. (2021), and CLIPs Radford et al. (2021). The complete list of models is available in Table 6 in the appendix. For brevity, we present results for DINO and ResNet models, and additional experiments for other models are provided in Appendix **??**.

For each dataset, we input the data into the models to obtain extracted features. A linear head was then trained on top of these features. In the case of ImageNet, the linear head was trained using 30 data points per class from the overall 30,000 training images. We designated 30,000 images from the ImageNet validation set as our **DebugSet** (30 per class), with the remaining 20,000 samples used for testing. The images had a resolution of $224 \times 224$, and the task was ImageNet classification.

The hyperparameters and other settings are detailed in Table 5 in the appendix.

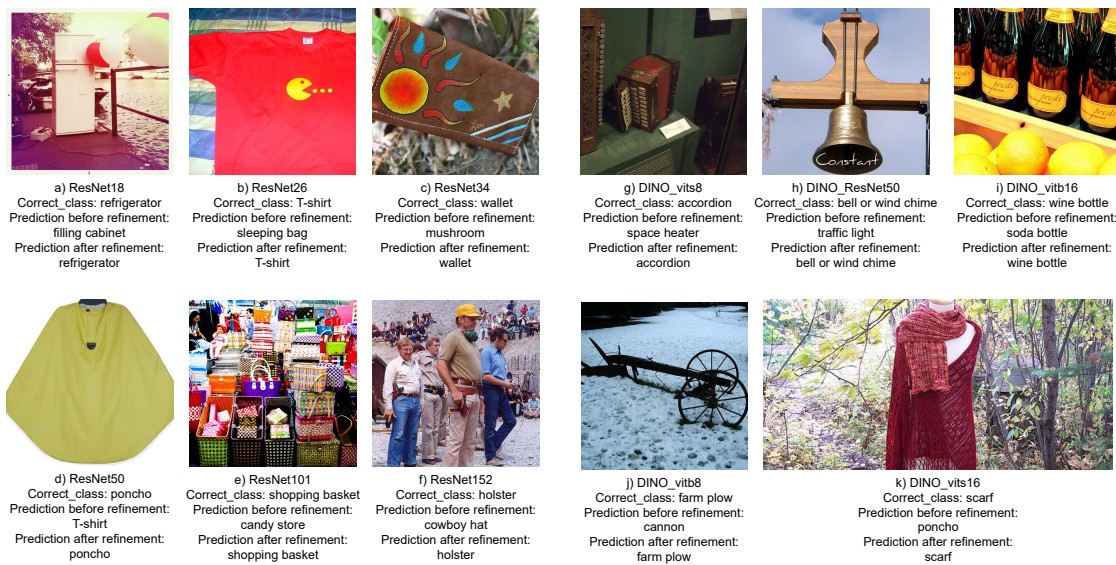

Figure 3: Some examples of failure modes of ResNets and DINOs

To address detected failures in the **DebugSet**, we split them in half. The first half, **DebugSeed**, was used for refinement. Uncommon backgrounds were generated using ChatGPT 3.5. The CLIP model used for selecting backgrounds for data points was ViT-B/32 CLIP. For synthetic data generation, we utilized Stable Diffusion V1-5 imported from the diffusers package von Platen et al..

## 5.2 Failure inspection

The initial stage of our framework involves analyzing how various models fail to classify objects in different datasets. To accomplish this, we use the CLIP model to identify backgrounds on which models struggle to classify objects. This results in captions that describe failures related to rare backgrounds. In the following stage, we examine these identified failures and explore how the generated captions help us to recover from them. We investigate results for both **individual and collective failure inspection**.

### 5.2.1 Individual Failure Inspection

In Figure 3, we show some instances where ResNet and DINO models have failed and see that these failures are due to wrong background association. In this Figure, the six images on the left (a-f) are examples of Resnets' failures, and the five images on the right (g-k) are failure modes' of DINO models. For example, image **c** shows *"a wallet in jungle"*, which can be regarded as an uncommon background for this object. As a result, the ResNet34 model is unable to classify it accurately and instead predicts a *"mushroom"* which is more likely to be found *"in jungle"*, especially under a plant, despite having no resemblance to the actual object in the image. Similarly, image **h** illustrates *"a bell or wind chime in sky"*, which is uncommon since *"bell"* is more likely to be seen with other backgrounds such as *"a door, a building or a wall"*. Therefore, the DINO_reset50 model mispredicts as *"traffic light"* because *"traffic light"* is more common to be seen *"in a sky background"*.

In general, our approach is capable of addressing failure scenarios originating from uncommon backgrounds of objects or any other spurious correlation (We provide another instance demonstrating how our framework can be applied to analyze another type of spurious correlation, such as color, in 5.3.2). Analyzing the relevant backgrounds allows us to readily understand the cause of such failure instances.

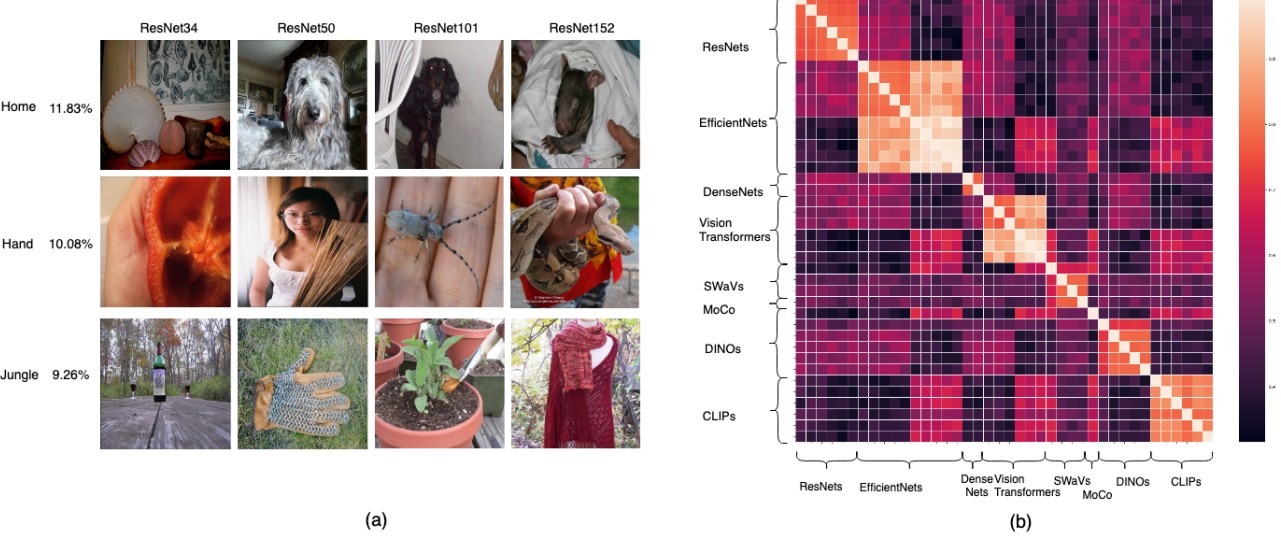

Figure 4: a) The most three common failure backgrounds in ResNets pretrained on ImageNet. The numbers in the image show the percentage of the shared failures that are related to the mentioned background. b) Comparing failures of all models **(Intersection/Union)**.We observe that models belonging to the same categories tend to exhibit more comparable failures.

### 5.2.2 Collective Failure Inspection

Within this section, we will compare the failure modes for all models to assess their alignment. We aim to determine the extent to which failures are consistent across models. While numerous studies have focused on analyzing similarities in the learning process and representations of different models, such as Raghu et al. (2021) that demonstrated the similarity between convolutional neural networks (convnets) and other convnets, as well as the similarity between vision transformers (ViTs) and other ViTs int the way they learn features, our focus is on investigating whether models also fail in similar ways. This will enable us to gain a deeper understanding of how to address the issue of failures in a more generalized manner without taking the specific model into consideration.

The failures of different models in various categories are compared in Figure 4 by computing the *intersection over union* of the failures. It can be observed that models within the same category fail in more similar samples. Typically, the failures between models from the same category are over 80% similar (e.g., CLIPs and EfficientNets). Among all 40 models, the intersection of failure modes is above 40%, indicating that models tend to fail in very similar ways, even with different architectures. Some examples of the shared failures relted to backgrounds can be found in 4. This will raise the question of "how to utilize this similarity in failures to enhance a group of models' performance?" which we will explore more in section 5.3.2.

It is pertinent to mention that Wiles et al. (2022) has also recognized patterns of consistencies in failures among models within the same category. However, we take a step further and leverage these consistencies to mitigate shared failure modes systematically.

## 5.3 Failure Mitigation

### 5.3.1 Individual Failure Mitigation

The outcomes from employing our framework are presented in Table 2. We only included the results for ResNets and DINOs, but we have results for other models (EfficientNets, DenseNets, ViTs, SWaVs, MoCo, and CLIPs) in the appendix. In Table 2, we constructed ***DebugSeed*** and ***DebugHeldout*** sets to yield zero accuracy for the model, as they are composed of model failures. Post refinement and utilizing ***DebugSeed***, we observe substantial improvements in ***DebugHeldout*** data that we did not use for refinement, ensuring an

| Models | | Accuracies | | | | | |
|--------|--------|------|------|------|------|------|------|
| Model category | model name | **Individual Refinement (ours)** | | | | Random Refinement | |
| | | Clean | Failure | Seed | **Heldout** | Seed | Heldout |
| ResNet | ResNet18 | 0.9891 | 0.0651 | 0.2636 | **0.2128** | 0.1134 | 0.1129 |
| | ResNet26 | 0.9783 | 0.0649 | 0.2856 | **0.228** | 0.09539 | 0.0904 |
| | ResNet34 | 0.9879 | 0.0781 | 0.3061 | **0.2531** | 0.09856 | 0.08615 |
| | ResNet50 | 0.9864 | 0.0607 | 0.3444 | **0.2717** | 0.1102 | 0.1072 |
| | ResNet101 | 0.9790 | 0.113 | 0.3574 | **0.2656** | 0.1132 | 0.1181 |
| | ResNet152 | 0.9901 | 0.0863 | 0.3609 | **0.2817** | 0.08207 | 0.08804 |
| DINO | ViT-S/8 | 0.9812 | 0.0536 | 0.3117 | **0.2494** | 0.1134 | 0.1129 |
| | ViT-S/16 | 0.9855 | 0.0462 | 0.2922 | **0.2379** | 0.09539 | 0.0904 |
| | ViT-B/8 | 0.9782 | 0.0655 | 0.3325 | **0.2518** | 0.09856 | 0.08615 |
| | ViT-B/16 | 0.9848 | 0.0388 | 0.3067 | **0.2477** | 0.1102 | 0.1072 |

Table 2: Our approach significantly outperforms Random Refinement, leading to an approximate $\sim 21\%$ improvement in accuracy across all models on the **DebugHeldout** dataset. This underscores the effectiveness of our method in rectifying errors attributed to incorrect background associations. Clean and Failure accuracies reflect the post-refinement performance of correctly and incorrectly classified data, respectively, which show that our method do not harm the clean accuracy by incorporating additional data. Prior to refinement, models exhibited zero accuracy on **DebugSeed** and **DebugHeldout** datasets.

unbiased evaluation. This improvement underscores that many failure modes stem from incorrect associations models make between objects and backgrounds. Some might argue that this gain results from the additional data. Thus, we present results for a baseline we term **Random refinement**. This baseline similarly uses **DebugSeed** and **DebugHeldout**, then generates synthetic data using only class names (prompts are structured as *"A photo of [class_name]"*). This comparison illustrates that the improvement of our method arises from considering background information. In the outcomes of **Random refinement**, the improvement over **DebugSeed** and **DebugHeldout** is roughly equivalent since no information from the background association of either set was utilized. **Random refinement** solely employs class names to generate data. However, this improvement is not on par with the gain achieved by incorporating background information when generating new data. We also include results for Color spurious associations on CIFAR-100 in table 3 to further support our claim. Results for iNaturalist-2018 can also be found in table 8.

It's worth noting that despite incorporating Stable Diffusion-generated data, which could be seen as out-of-distribution samples, a positive impact on model performance remains. This is primarily attributed to the parameter **lambda** that controls the contribution of the generated images in our training process. The influence of this parameter is depicted in Figure 5.

Another crucial hyper-parameter is the number (#) of generated synthetic data per class. The effect of this hyper-parameter, denoted as **k**, is illustrated in Figure 5.

The improvement observed in **DebugHeldout** surpasses $\sim 21\%$ for all models, highlighting the tendency of models to fail in associating backgrounds with objects and utilizing this association to predict objects, neglecting object-specific features. This can be contrasted with the accuracy gain achieved by the **Random refinement** baseline, which is significantly smaller than our method. We also include results of comparing our framework with other methods such as Jain et al. (2023); Yun et al. (2019); Singla et al. (2024) in section B in appendix. The results show the superiority of our spurious correlation identification and mitigation framework comparing to other related work.

### 5.3.2 Collective Failure Mitigation

As discussed in section 5.2.2, since models from the same categories have very similar failures, we have considered the possibility of using a single set of generated data, called **Collective_DebugTrain**, to refine

| Models | Accuracies | | | | | | |
|--------|-----------|---|---|---|---|---|---|
| Model category | model name | **Individual Refinement** (ours) | | | | Random Refinement | |
| | | Clean | Failure | Seed | Heldout | Seed | Heldout |
| ResNet | ResNet18 | 0.9892 | 0.1305 | 0.3323 | **0.2818** | 0.1901 | 0.1878 |
| | ResNet34 | 0.9952 | 0.1560 | 0.3586 | **0.2923** | 0.1902 | 0.1947 |
| | ResNet50 | 0.9931 | 0.1394 | 0.3614 | **0.3008** | 0.1971 | 0.1858 |
| | ResNet101 | 0.9991 | 0.169 | 0.3877 | **0.3249** | 0.2001 | 0.1930 |
| | ResNet152 | 0.9943 | 0.1525 | 0.3842 | **0.3184** | 0.2139 | 0.2152 |

Table 3: Accuracy of our method compared to the Random refinement on CIFAR-100 considering color spurious correlations. The prompt used in the above table to input ChatGPT is "What is an uncommon color that <class_name> may possess?", and the prompt for generating more data with Stable Diffusion is "A photo of a <color> <class_name>.". Note that the accuracy of models on **DebugSeed** and **DebugHeldout** was zero before refinement. After applying our method, we gain above $\sim 28\%$ improvements in accuracies for all models, showcasing that more than $\sim 28\%$ of model errors in the heldout set come from wrong color associations.

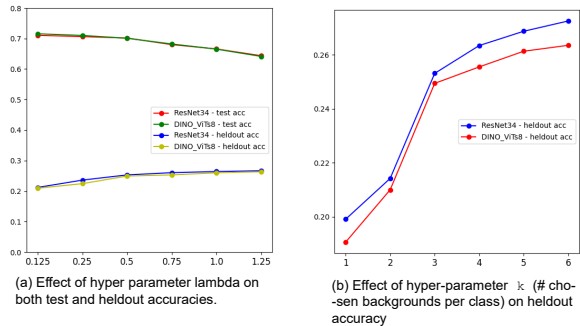

(a) Effect of hyper parameter lambda on both test and heldout accuracies.

(b) Effect of hyper-parameter $k$ (# cho--sen backgrounds per class) on heldout accuracy

Figure 5: a) As the value of **lambda** increases, the accuracy on **DebugHeldout** improves while the accuracy on the test set (containing both clean and failure examples) decreases, meaning increasing lambda by a certain value will harm the clean accuracy. However, there is a specific point **(0.5)** on the plot where the accuracy of the models on both the test and heldout sets stabilizes. b) Increasing the number of chosen backgrounds per class enhances the accuracy on the **DebugHeldout**. Considering the high cost of generating additional data, we opt for **k = 3**, where the plot exhibits a significant slope, and the few-shot generation of additional data will help in accuracy improvement.

all models within the same categories. To achieve this, we have devised two different settings: 1) we get the failure modes of all models in the same category (e.g. ResNets), and we select **k** samples from all the failures in each class. Therefore, background failures that occurred more have a higher probability of being chosen for the **Collective_DebugTrain**. We then use this data to refine individual models in this category. 2) We get the failure modes of only one of the models in a category and then use this to refine all models. This approach is more efficient in terms of time and memory, as it requires running only one model per category. The results for this experiment are shown in table 4. Based on our observations in section 5.2.2, having the same **DebugTrain** data for refinement (**Collective_DebugTrain**), improves the accuracies among all models in the same category. This approach offers greater efficiency as it eliminates the need to generate **DebugTrain** data for each individual model. Consequently, it saves us both time and memory that would otherwise be required for storing such data.

In an overview, the **collective refinement** approach showcases the capability to resolve above **75%** of failures corrected by individual refinement 6.

| Models | Accuracies | | | | | | | |
|--------|------------|---|---|---|---|---|---|---|
| Model category | model name | Before debugging | **Collective refinement-type 1** (ours) | | | **Collective refinement-type 2** (ours) | | |
| | | Test | Test | seed | **heldout** | Test | seed | **heldout** |
| ResNet | ResNet18 | 0.6236 | 0.6364 | 0.2291 | **0.2078** | 0.6413 | 0.2636 | **0.2128** |
| | ResNet26 | 0.6593 | 0.6655 | 0.2396 | **0.2192** | 0.6669 | 0.2185 | **0.1853** |
| | ResNet34 | 0.7017 | 0.7149 | 0.2312 | **0.2188** | 0.7135 | 0.2391 | **0.2178** |
| | ResNet50 | 0.7631 | 0.7644 | 0.2419 | **0.2217** | 0.7641 | 0.2325 | **0.2105** |
| | ResNet101 | 0.796 | 0.8001 | 0.2474 | **0.2226** | 0.7999 | 0.2244 | **0.2045** |
| | ResNet152 | 0.816 | 0.8182 | 0.2509 | **0.2317** | 0.8188 | 0.2253 | **0.2081** |
| DINO | ViT-S/8 | 0.6977 | 0.70001 | 0.2729 | **0.2488** | 0.70008 | 0.3117 | **0.2494** |
| | ViT-S/16 | 0.649 | 0.6522 | 0.2673 | **0.2394** | 0.6504 | 0.2563 | **0.2267** |
| | ViT-B/8 | 0.7101 | 0.7125 | 0.2913 | **0.2509** | 0.7117 | 0.2903 | **0.2540** |
| | ViT-B/16 | 0.6832 | 0.6840 | 0.2985 | **0.2467** | 0.6833 | 0.2855 | **0.2488** |

Table 4: Collective refinement results.

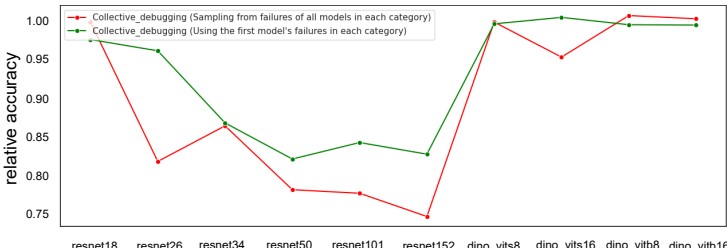

Figure 6: Comparison of resolved failures between collective refinement and individual refinement as a percentage. Relative accuracy is the ratio of the collective refinement's accuracy over individual refinement's accuracy on each specific model. Our **collective refinement** method is able to resolve more than 75% of individual model's failures.

# 6 Conclusion

In this project, we have developed a technique to identify failure modes by focusing on a specific category of spurious correlations. We then leverage these detected failures to generate additional samples, allowing the model to learn from and address its shortcomings. We have illustrated the resemblance of failures within a particular model category, highlighting that models with the same architecture share more similar failures. Exploiting this insight, we have devised a method to alleviate failures across all models in a category using a single set of generated data based on the failures of just one model in that category. Our results indicate that **collective refinement** approach can resolve over 75% of failures addressed through **individual refinement** efforts. Our framework empowers users to select the spurious correlation to identify and mitigate, facilitating the simultaneous refinement of a subset of models with a single (small) auxiliary set of additional data, thereby saving both time and resource.

# 7 Discussion

**Failure modes of Stable Diffusion**: Prior work have investigated failure modes of diffusion-based models. Liu et al. (2023) proposes SAGE, an adversarial search method that explores failure modes in Text-Guided Diffusion Models (TDMs), revealing issues like generating inaccurate images and identifying misalignments between latent and prompt spaces. Some other work explore the problem of compositionality in Text-Guided generative models by using manually crafted prompts, including Gokhale et al. (2022); Marcus et al. (2022); Conwell & Ullman (2023). Another line of work have studied biases in generative models including societal

biases Luccioni et al. (2023); Saravanan et al. (2023) and gender biases Wu et al. (2023). While these studies highlight significant challenges and shortcomings in text-to-image generative models, it is important to note that, for the practical purpose of automatic generation, these models remain a widely adopted approach, and are currently represent the most effective solution available for automated content generation.

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

## A  Appendix

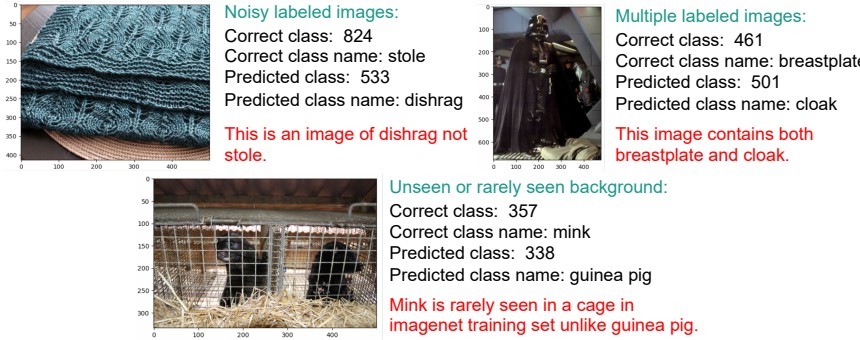

Figure 7: examples of 3 most common failure modes of deep learning models

| Parameter | Value |
|---|---|
| Leearning rate | 0.2 |
| Epochs | 1000 |
| Momentum | 0.9 |
| Weight decay | 0.0005 |
| # Chosen common BGs | 3 |
| *lambda* | 0.5 |

Table 5: Shared parameters among all dataset.

| Models | | |
|---|---|---|
| Model_category | Model_name | |
| ResNet | ResNet18
ResNet50 | ResNet26
ResNet101 | ResNet34
ResNet152 |
| EfficientNet | efficientnet_b0
efficientnet_b3
efficientnet_b6
efficientnet_l2 | efficientnet_b1
efficientnet_b4
efficientnet_b7 | efficientnet_b2
efficientnet_b5
efficientnet_b8 |
| DenseNet | densenet121 | densenet161 | |
| ViT | vit_base_patch16_224
vit_large_patch32_224 | vit_base_patch32_224
vit_base_resnet26d_224 | vit_large_patch16_224
vit_base_resnet50d_224 |
| SWaV | resnet50
resnet50w5 | resnet50w2 | resnet50w4 |
| MoCo | moco_v2_800ep | | |
| DINO | dino_resnet50
dino_vits16 | dino_vitb16
dino_vits8 | dino_vitb8 |
| CLIP | ViT-B32
ViT-L14 | RN50 | RN101 |

Table 6: List of models we tested our refinement framework on.

| Models | | Accuracies | | | | | | |
|---|---|---|---|---|---|---|---|---|
| Model category | model name | before refinement | **Individual refinement (ours)** | | | Random refinement | | |
| | | Test | Test | seed | **heldout** | Test | seed | heldout |
| EfficientNet | b0 | 0.715 | 0.7198 | 0.2034 | **0.1842** | 0.7134 | 0.0894 | 0.0883 |
| | b1 | 0.7373 | 0.74415 | 0.2154 | **0.1885** | 0.7399 | 0.1003 | 0.1011 |
| | b2 | 0.7525 | 0.7591 | 0.2283 | **0.1909** | 0.7448 | 0.0957 | 0.0942 |
| | b3 | 0.7634 | 0.7730 | 0.2318 | **0.1991** | 0.7669 | 0.1066 | 0.1010 |
| | b4 | 0.7701 | 0.7780 | 0.2398 | **0.2068** | 0.7719 | 0.0955 | 0.0960 |
| | b5 | 0.7821 | 0.7819 | 0.2405 | **0.2055** | 0.7761 | 0.0893 | 0.0915 |
| | b6 | 0.7884 | 0.7886 | 0.2561 | **0.2083** | 0.7863 | 0.0941 | 0.0914 |
| | b7 | 0.7895 | 0.7903 | 0.2600 | **0.2126** | 0.7898 | 0.0951 | 0.0972 |
| | b8 | 0.7928 | 0.7951 | 0.2653 | **0.2147** | 0.7932 | 0.0972 | 0.0934 |
| DenseNet | 121 | 0.6792 | 0.6869 | 0.2138 | **0.1592** | 0.6773 | 0.0651 | 0.0664 |
| | 161 | 0.7254 | 0.7332 | 0.2418 | **0.1833** | 0.7249 | 0.0779 | 0.0771 |
| ViT | base_patch16_224 | 0.739 | 0.7477 | 0.2501 | **0.2193** | 0.7399 | 0.1047 | 0.1044 |
| | base_patch32_224 | 0.7456 | 0.7493 | 0.2574 | **0.2199** | 0.7469 | 0.1078 | 0.1072 |
| | large_patch16_224 | 0.7493 | 0.7539 | 0.2644 | **0.2263** | 0.7468 | 0.0952 | 0.0957 |
| | large_patch32_224 | 0.7535 | 0.7545 | 0.2674 | **0.2274** | 0.7553 | 0.1023 | 0.1041 |
| SWaV | resnet50 | 0.4254 | 0.4384 | 0.1403 | **0.1274** | 0.4267 | 0.0662 | 0.0624 |
| | resnet50w2 | 0.4317 | 0.4328 | 0.1583 | **0.1294** | 0.4319 | 0.0683 | 0.0652 |
| | resnet50w4 | 0.4402 | 0.4477 | 0.1592 | **0.1304** | 0.4416 | 0.0672 | 0.0617 |
| | resnet50w5 | 0.4526 | 0.4589 | 0.1633 | **0.1363** | 0.4552 | 0.0696 | 0.0703 |
| MoCo | v2_800ep | 0.6931 | 0.6946 | 0.2041 | **0.1584** | 0.6937 | 0.0943 | 0.0917 |
| CLIP | ViT-B32 | 0.5388 | 0.5582 | 0.1794 | **0.1635** | 0.5407 | 0.0776 | 0.0763 |
| | ViT-L14 | 0.7427 | 0.7694 | 0.2174 | **0.2068** | 0.7505 | 0.0893 | 0.0981 |
| | RN50 | 0.5928 | 0.6129 | 0.1980 | **0.1833** | 0.6004 | 0.0964 | 0.0946 |
| | RN101 | 0.7532 | 0.7751 | 0.2566 | **0.2142** | 0.7570 | 0.1084 | 0.1115 |

Table 7: Accuracy of our method comparing to **Random refinement**. Note that the accuracy of models on *DebugSeed* and *DebugHeldout* was zero before refinement.

| Models | | Accuracies | | | | | | |
|---|---|---|---|---|---|---|---|---|
| Model category | model name | Accuracy before debugging | **Accuracy of Individual Debugging (ours)** | | | Accuracy of Random debugging | | |
| | | Test | Test | seed | **heldout** | Test | seed | heldout |
| ResNet | ResNet50 | 0.7581 | 0.7852 | 0.3612 | **0.3082** | 0.7614 | 0.1348 | 0.1352 |
| | ResNet101 | 0.7863 | 0.8148 | 0.3910 | **0.3173** | 0.7899 | 0.1377 | 0.1384 |
| | ResNet152 | 0.7996 | 0.8252 | 0.3916 | **0.3194** | 0.8015 | 0.1485 | 0.1463 |

Table 8: Accuracy of our method compared to the **Random refinement** on iNaturalist-2018 considering background spurious correlations. Note that the accuracy of models on *DebugSeed* and *DebugHeldout* was zero before refinement.

To better compare the superiority of using ChatGPT for uncommon spurious correlation suggestion, we include the accuracy improvement on ImageNet for mitigating background associations when using a pre-defined set of backhgrounds which contain ["in a blur background", "on a leaf", "in water", "on soil", "on a plate", "in a hand", "on a sofa", "in garden", "in jungle", "in cave", "on snow", "in a plane background", "in yard", "outdoor", "in shore", "in the sky", "indoor", "on a wall", "on a tree", "on a table", "in a street", "on a rock", "in a airplane", "in cage", "with sun", "in a mountain", "in metro", "on grass", "in shelf", "on rails",

| Models | | Accuracies | |
|---|---|---|---|
| Model category | model name | **Accuracy of Individual Debugging** (ours) | |
| | | seed | **heldout** |
| ResNet | ResNet18 | 0.2260 | **0.1690** |
| | ResNet26 | 0.2295 | **0.1751** |
| | ResNet34 | 0.2294 | **0.1788** |
| | ResNet50 | 0.2354 | **0.1806** |
| | ResNet101 | 0.2384 | **0.1847** |
| | ResNet152 | 0.2402 | **0.1857** |
| DINO | ViT-S/8 | 0.2310 | **0.2073** |
| | ViT-S/16 | 0.2231 | **0.2044** |
| | ViT-B/8 | 0.2492 | **0.2271** |
| | ViT-B/16 | 0.2359 | **0.2118** |

Table 9: Accuracy of our method when using a set of pre-defined backgrounds to mitigate background spurious correlations in ImageNet dataset. Results show the superiority of using ChatGPT for rare background suggestion.

"with a person", "on bed", "in a playground", "in a kitchen", "on floor", "on glass", "on wood", "at a party", "in a wardrobe", "in a restaurant", "in a bucket"]. The results for this experiment can be seen in table 9.

## B  Comparing to baselines

### B.1  CutMix Yun et al. (2019)

**Method**: The CutMix augmentation strategy is proposed to improve regional dropout methods for training convolutional neural network classifiers. Instead of removing informative pixels with patches of black pixels or random noise, CutMix involves cutting and pasting patches among training images, with ground truth labels mixed proportionally to the patch area. This approach efficiently utilizes training pixels while retaining the regularization effect of regional dropout.

**Setting**: We employ the implementation provided by the authors of the paper. The pretrained models were trained using the below configuration:

| Parameter | Value |
|---|---|
| net_type | resnet |
| dataset | imagenet |
| batch_size | 256 |
| lr | 0.1 |
| depth | 50 |
| epochs | 300 |
| expname | ResNet50 |
| j | 40 |
| beta | 1.0 |
| cutmix_prob | 1.0 |
| verbose | No |

Table 10: CutMix experiment configuration on ImageNet (e.g., ResNet50)

## B.2 Jain et al. (2023)

**Method**: They automatically distill failure modes in machine learning models to offer a global understanding of datasets. The method involves representing failure modes as directions in a feature space by training linear classifiers. This allows for the automatic detection, interpretation, and intervention of model failures. Shared vision/language embeddings like CLIP are leveraged to ensure consistency.

**Setting**: We use the official GitHub implementation provided by the paper. All the settings, including the percentage of validation set and ***DebugSet***, are the same as ours. We compare their result with ours on ***DebugHeldout*** in table 11.

## B.3 DCD Singla et al. (2024)

**Method**: DCD addresses model failures in deep neural networks, particularly in scenarios where the training set inadequately covers diverse deployment settings. Focusing on image classification, DCD leverages a small set of samples from an error distribution (Esample) and a large pool of weakly labeled data (F). The framework systematically improves model performance on the error distribution while maintaining accuracy on the original test set. DCD strategically selects visually similar images from F by using the $l2$ distance in the penultimate layer activations of various models.

**Comparison**: Since the paper's code is not available, the frameworks are compared based on their methods to find and mitigate failures.

- The identified failures lack interpretability, a crucial aspect for sanity checks and gaining deeper insights into the models' behaviors.

- The method involves a time-consuming manual process, including the selection of a subset of classes with low accuracy, gathering synsets, and performing a Flickr search for synonyms. This results in collecting a substantial number of image URLs (e.g., 952,951 across 160 ImageNet classes) and adds complexity by removing common URLs across classes. In contrast, our method only adds **3** images per class, significantly reducing the additional data burden.

In table 11, we compare our results with Jain et al. (2023); Yun et al. (2019). Note that the final models are all tested on the same ***DebugHeldout*** for a fair comparison.

| Models | | Accuracies on ***DebugHeldout*** | | |
|---|---|---|---|---|
| Model Category | model name | **Ours** | Jain et al. (2023) | Yun et al. (2019) CutMix |
| ResNet | ResNet18 | **0.2128** | 0.0923 | 0.0642 |
| | ResNet26 | **0.228** | 0.1023 | 0.0527 |
| | ResNet34 | **0.2531** | 0.0839 | 0.0655 |
| | ResNet50 | **0.2717** | 0.1185 | 0.0738 |
| | ResNet101 | **0.2656** | 0.1147 | 0.0870 |
| | ResNet152 | **0.2817** | 0.1275 | 0.0941 |

Table 11: Comparison of our framework with some baselines.

