# OpenReview forum: "Identifying and Mitigating Model Failures through Few-shot CLIP-aided Diffusion Generation"
_TMLR — Rejected by TMLR_

### Review · Reviewer_KxqL · 2024-01-13

**Summary Of Contributions:**

The paper proposes an automated framework to identify and mitigate model failures from spurious correlations like wrong background associations, without human intervention. It detects failure modes using CLIP and ChatGPT on incorrectly predicted examples, and generates synthetic debug data with Stable Diffusion. Only the linear classification heads of models are retrained on the original and generated data. Experiments on 40 models for ImageNet, CIFAR-10 and CIFAR-100 show over 21% average accuracy gain on held-out data from interpreting failures as uncommon backgrounds. Models within the same category tend to exhibit similar failure modes, enabling collective debugging where a unified synthetic debug set improves multiple models efficiently. In summary, this end-to-end framework automatically interprets, generates targeted data for, and mitigates various model failures caused by spurious correlations, demonstrating state-of-the-art performance on multiple datasets and architectures.

**Audience:**

Yes

**Claims And Evidence:**

Yes

**Requested Changes:**

Look at the above shortcomings.
By the way, I found the paper writing contains some typos, for examples, the lower case or upper case. Please modify.

**Strengths And Weaknesses:**

For the strengths, it has following points:
1. The generalizability. This framework can be applied to different scenarios, instead of only the background field as the authors presented.
2. The framework is interpretable and can help the understanding of the wrongly predicted cases.

As for the weaknesses, I have several concerns that should be addressed:
1. In terms of the framework, it is automatic, which is good. However, it requires multiple different model assistance, which may not be so applied. The cost and also the model performance (assistant models) would seriously impact the final result.
2. In terms of the background or other aspects, one by one detection and usage of the framework looks to be quite unrealistic. It should be clear that the wrong prediction is not caused by only one reason, therefore, the authors are encouraged to make improvement.
3. In terms of the utilization of ChatGPT, the problem is over-reliance on prompts - The success of the method relies completely on the ability to specify failure modes through natural language prompts. This may not always be possible, limiting the technique's applicability.
4. Lack of theoretical guarantees - There is no analysis proving the generated data and debugging process will consistently improve models. The framework makes assumptions but provides no theoretical backing.

---

> ### Author Response · Authors · 2024-01-25
>
> We appreciate the reviewer for providing valuable comments on our paper.
>
> > In terms of the framework, it is automatic, which is good. However, it requires multiple ...
>
> We acknowledge that the use of multiple models may make the final result depend heavily on each model’s performance. The problem of failure mode (particularly spurious correlations) inspection and mitigation has been present since the very beginning, and as of our knowledge, there has been no work that solves this issue 100%. The paper here attempts to address this problem with the use of the most recent and powerful models to both achieve good results and make the whole framework more and more automatic. Although the three models mentioned—CLIP, Stable Diffusion, and ChatGPT—are not 100% accurate, they are the only available options trained on massive training data, and as presented in experiments these assistant models improve the performance without human-intervention, and they can be deployed on real-world datasets as well since they are fast and automatic and need no supervision.
>
> Regarding the cost of using these assistant models, since the proportion of the held-out set used to mitigate failure modes is very small compared to all the training data, the cost of using these models is negligible compared to their ability to automatically refine the failure modes. Moreover, since the final step of improving the models’ performance is to only retrain the linear head, the overall cost of applying the framework is small. It can even be run on Google Colab without the need for powerful GPUs.
>
> > In terms of the background or other aspects, one by one detection and usage ...
>
> As the reviewer mentioning here, the failure modes of a model can arise from many different reasons. There are many works (cited in the related work section) that try to improve the performance of a model on a dataset without considering any specific failure case (for example background, color association). Although in most cases, they successfully improve the performance, one big problem is that most of the proposed methods are not interpretable. In this paper, we focused on the interpretability of the framework as well to increase the knowledge about failures that a model makes. This will give us insight on how to refine the model with respect to different failure cases. In addition, the detection of failure modes with respect to one aspect has another benefit which is the user can select the most important aspect to refine based on their dataset. For example if their dataset is “Natural-Color Dataset” [A], since the color of different vegetables and fruits can cause a prior bias for the model, then the user may want to mitigate failure modes that are related to color. We add the experiments of color spurious correlation on CIFAR-100 here to better show the applicability of our framework.
>
>
> | Models |  |  |  |  Accuracies |  |  |  |  |  |
> | :---: | :---: | :---: | :---: | :---: | :---: | :---: | :---: | :---: | :---: |
> | Model category | model name | Accuracy  before debugging |  | Accuracy of Individual Debugging (ours) |  |  | Accuracy of Random debugging |  |  |
> |  |  | Test | Test | seed | heldout |  | Test | seed | heldout |
> |  | resnet18 | 0.7551 | 0.7789 | 0.3323 | $\mathbf{0 . 2 8 1 8}$ |  |  0.7591 | 0.1901  |  0.1878 |
> |  | resnet34 | 0.7666 | 0.7994 | 0.3586 | $\mathbf{0 . 2 9 2 3}$ |  | 0.7715 | 0.1902  |  0.1947 |
> | ResNets | resnet50 | 0.7730 | 0.7993 | 0.3614 | $\mathbf{0 . 3 0 0 8}$ |  | 0.7715  |  0.1971 | 0.1858 |
> |  | ResNet101 | 0.7748 | 0.8123 | 0.3877 | $\mathbf{0 . 3 2 4 9}$ |  | 0.7811 | 0.2001  |  0.1930 |
> |  | resnet152 | 0.7769 | 0.8065 | 0.3842 | $\mathbf{0 . 3 1 8 4}$ |  | 0.7797 | 0.2139  |  0.2152 |
>
> The prompt used in the above table to input ChatGPT is "What is an uncommon color that <class_name> may possess?", and the prompt for generating more data with Stable diffusion is "A photo of a <color> <class_name>"
>
> The answers to the rest of the reviewer's comments can be found in the next comment.

---

> ### Author Response · Authors · 2024-01-25
>
> > In terms of the utilization of ChatGPT, the problem is over-reliance on prompts ...
>
> We acknowledge the reviewer's point that the current approach relies on natural language prompts to specify failure modes. However, for the sake of automatically generating failure modes, we need to deploy a large language model which is quite common in this context (e.g. https://arxiv.org/pdf/2310.00164.pdf). The effectiveness of using ChatGPT, besides avoiding human labor, can be seen in Table 2, in which we demonstrate that using the spurious correlations suggested by ChatGPT along with captioning capabilities of CLIP will result in much better improvement comparing to when we do not use any of this information and rely solely on class names.
> Moreover, we have also tested our framework with a set of predefined spurious associations that we collected based on the knowledge we have regarding a dataset, the results can be seen here:
>
> | Models |  | Accuracies |  |
> | :---: | :---: | :---: | :---: |
> | Model category | Model name |  |  Accuracy of Individual Debugging|
> |  |  |  seed | heldout |
> |  | resnet18 | 0.2260 | $\mathbf{0 . 1 6 9 0}$ |
> |  | resnet26 | 0.2295 | $\mathbf{0 . 1 7 5 1}$ |
> | ResNets | resnet34 | 0.2294 | $\mathbf{0 . 1 7 8 8}$ |
> |  | resnet50 | 0.2354 | $\mathbf{0 . 1 8 0 6}$ |
> |  | resnet101 | 0.2384 | $\mathbf{0 . 1 8 4 7}$ |
> |  | resnet152 | 0.2402 | $\mathbf{0 . 1 8 5 7}$ |
> |  | ViTs8 | 0.2310 | $\mathbf{0 . 2 0 7 3}$ |
> |  | ViTs16 | 0.2231 | $\mathbf{0 . 2 0 4 4}$ |
> | DINO | ViTb8 | 0.2492 | $\mathbf{0 . 2 2 7 1}$ |
> |  | ViTb16 | 0.2359 | $\mathbf{0 . 2 1 1 8}$ |
> |  | resnet50 | 0.1904 | $\mathbf{0 . 1 7 4 2}$ |
>
> The above table shows the result for when we choose a set of 40 predefined backgrounds with respect to the knowledge we have from the ImageNet dataset. These backgrounds are ["in a blur background", "on a leaf", "in water", "on soil", "on a plate", "in a hand", "on a sofa", "in garden", "in jungle", "in cave", "on snow", "in a plane background", "in yard", "outdoor", "in shore", "in the sky", "indoor", "on a wall", "on a tree", "on a table", "in a street", "on a rock", "in a airplane", "in cage", "with sun", "in a mountain", "in metro", "on grass", "in shelf", "on rails", "with a person", "on bed", "in a playground", "in a kitchen", "on floor", "on glass", "on wood", "at a party", "in a wardrobe", "in a restaurant", "in a bucket"].
>
> The improvement we observe from using ChatGPT is more than that of a predefined set which is suggested by a human. This is understandable since Large Language Models like ChatGPT have been trained on massive amounts of data and they have more knowledge regarding this manner. This along with the automatic generation of spurious correlations is the reason we preferred ChatGPT, with all its shortcomings, over relying on humans.
>
> > Lack of theoretical guarantees ...
>
> Thank you for bringing this up. This is true that we had some assumptions and we demonstrated our claims via experimental setups not theoretically. We observed that many of the failures of a model is due to different spurious correlations (e.g. Figure 3 that shows the models rely heavily on spurious correlations for making a decision.). The experiments also show this. Both random_debugging and our framework(individual and collective debugging) show some improvement when we add new generated data based on the detected failures, and improvement for individual and collective debugging is much more than random_debugging since we take spurious correlations into account. So our method is mostly based on known facts about the importance of spurious correlations and improvement by increasing training data.
>
> Moreover, about collective debugging which as of our best knowledge is the first work that improves the performance of a group of models by detecting failures of only one model, there is some work that shows models with same architectures tend to learn in more similar ways. Such as https://arxiv.org/abs/2108.08810 which proves their point by extensive experimental setups and other works such as https://arxiv.org/abs/2105.07581 shows that ResNets rely more on high-frequency discriminative features while ViT learns from both low and high frequencies with more focus on low frequencies in a more holistic manner. Therefore, they focus on different parts of their input and it's understandable that they may make more similar failures with other models from their own category.
>
> [A] Anwar, Saeed, et al. "Image colorization: A survey and dataset." arXiv preprint arXiv:2008.10774 (2020).

---

### Review · Reviewer_WmYW · 2024-01-15

**Summary Of Contributions:**

This paper investigates the failure modes of supervised deep learning models on ImageNet, with a primary focus on spurious correlations. The authors propose an end-to-end framework that utilizes foundation models, including ChatGPT, CLIP, and Stable Diffusion, to automatically identify and mitigate these failures. Specifically, the framework employs ChatGPT and CLIP to identify possible backgrounds contributing to the failures. It then uses diffusion models to generate auxiliary datasets for retraining the classifier layer. Experiments conducted on ImageNet, involving multiple types of models, demonstrate the effectiveness of the proposed framework. Additionally, the authors make an interesting observation regarding similar failures within the same category, suggesting the potential for efficient failure mitigation among models within the same group.

**Audience:**

Yes

**Claims And Evidence:**

Yes

**Requested Changes:**

In addition to the concerns outlined in the weaknesses section, I offer the following suggestions for improvement:

1. Enhance clarity by incorporating a figure that provides a visual example of applying the framework to identify and mitigate failures. Additionally, consider including pseudo-codes or an algorithm to enhance the understanding of the proposed method.

2. Clarify whether the observation of similar failures is an original contribution of this paper or if it has been identified in prior studies. If the observation is original, explicitly highlight it in the summarized contributions to underscore its significance.

**Strengths And Weaknesses:**

***[Strengths]***
1. The investigation of failure modes holds great significance for various deep-learning applications.

2. The framework is straightforward, simple, and well-founded.

3. The observation of similar failures within the same model group is insightful.



***[Weaknesses]***
1. The presentation lacks clarity due to significant missing information. While the number of selected backgrounds per class (k in Figure 1) is provided, details about the suggested uncommon backgrounds by ChatGPT and the number of images generated by Stable Diffusion remain undisclosed. Additionally, the claimed applicability of the framework to uncommon colors lacks demonstrations, and results for two CIFAR datasets are notably absent.

2. The impact of the paper is somewhat limited as it solely focuses on background issues. A more comprehensive understanding of failure modes, as presented in a recent study on remaining mistakes in ImageNet [A], encompasses various reasons. It remains unclear how the proposed framework addresses general failure modes.

3. The effectiveness of the framework appears to be inconsistent. While improvements are evident in mitigating failures related to uncommon backgrounds, as specified by its focused approach, results in Table 1 and Table 6 (Appendix) show only marginal enhancements or even reduced testing accuracy. It raises questions about potential misclassifications of correct samples by the updated classifier layer.

4. The experiments are insufficient, primarily relying on ImageNet results. As ImageNet datasets predominantly emphasize objects and maintain class balance, the proposed framework's effectiveness and generalization should be demonstrated on other realistic datasets, such as Places-205 and iNaturalist.

***[A]*** When does dough become a bagel? analyzing the remaining mistakes on imagenet. NeurIPS 2022

---

> ### Author Response · Authors · 2024-01-25
>
> We thank the reviewer for providing valuable comments on our paper.
>
> > The presentation lacks clarity due to significant ...
>
> We thank the reviewer for bringing this into our attention. To clarify the process, at first the uncommon backgrounds for each class will be suggested by ChatGPT, and then we input each instance from held-out set (The set we use to debug the model) to CLIP and it will choose the background for each of these failures from the set of uncommon backgrounds from the instance’s class. Then for each class, we select the top k backgrounds that are being chosen by CLIP most frequently. We hope this description here clarifies your concern.
> Upon the next evaluation we choose to provide the results for color spurious association" on CIFAR100, and we also add this to our dataset. The results are here:
>
> | Models |  |  |  |  Accuracies |  |  |  |  |  |
> | :---: | :---: | :---: | :---: | :---: | :---: | :---: | :---: | :---: | :---: |
> | Model category | model name | Accuracy  before debugging |  | Accuracy of Individual Debugging (ours) |  |  | Accuracy of Random debugging |  |  |
> |  |  | Test | Test | seed | heldout |  | Test | seed | heldout |
> |  | resnet18 | 0.7551 | 0.7789 | 0.3323 | $\mathbf{0 . 2 8 1 8}$ |  |  0.7591 | 0.1901  |  0.1878 |
> |  | resnet34 | 0.7666 | 0.7994 | 0.3586 | $\mathbf{0 . 2 9 2 3}$ |  | 0.7715 | 0.1902  |  0.1947 |
> | ResNets | resnet50 | 0.7730 | 0.7993 | 0.3614 | $\mathbf{0 . 3 0 0 8}$ |  | 0.7715  |  0.1971 | 0.1858 |
> |  | ResNet101 | 0.7748 | 0.8123 | 0.3877 | $\mathbf{0 . 3 2 4 9}$ |  | 0.7811 | 0.2001  |  0.1930 |
> |  | resnet152 | 0.7769 | 0.8065 | 0.3842 | $\mathbf{0 . 3 1 8 4}$ |  | 0.7797 | 0.2139  |  0.2152 |
>
> The prompt used in the above table to input ChatGPT is "What is an uncommon color that <class_name> may possess?", and the prompt for generating more data with Stable diffusion is "A photo of a <color> <class_name>"
>
> Results for iNaturalist-2018:
>
> | Models |  |  |  |  Accuracies |  |  |  |  |  |
> | :---: | :---: | :---: | :---: | :---: | :---: | :---: | :---: | :---: | :---: |
> | Model category | model name | Accuracy before debugging |  | Accuracy of Individual Debugging (ours) |  |  | Accuracy of Random debugging |  |  |  |
> |  |  | Test | Test | seed | heldout | Test | seed | heldout |  |
> |  | resnet50 | 0.7581 | 0.7852 | 0.3612 | $\mathbf{0 . 3 0 8 2}$ | 0.7614 | 0.1348 | 0.1352 |  |
> | ResNet | resnet101 | 0.7863 | 0.8148 | 0.3910 | $\mathbf{0 . 3 1 7 3}$ | 0.7899 | 0.1377 | 0.1384 |  |
> |  | resnet152 | 0.7996 | 0.8252 | 0.3916 | $\mathbf{0 . 3 1 9 4}$ | 0.8015 | 0.1485 | 0.1463 |  |
>
> *Accuracy of our method compared to the Random_debugging on iNaturalist-2018 considering
> background spurious correlations. Note that the accuracy of models on debug_seed and debug_heldout was
> zero before debugging.
>
> > The impact of the paper is somewhat ...
>
> Thanks for mentioning this important point. It is quite clear that the failure modes of a model on a dataset can be due to many different reasons. There are many works (cited in the related work section) that try to improve the performance of a model on a dataset without considering any specific failure case (for example background, color association). Although in most cases, they successfully improve the performance, one big problem is that most of the proposed methods are not interpretable. In this paper, we focused on the interpretability of the framework as well to increase the knowledge about failures that a model makes. This will give us insight on how to refine the model with respect to different failure cases. In addition, the detection of failure modes with respect to one aspect has another benefit, which is the user can select the most important aspect to refine based on their dataset without paying extra cost to detect and mitigate all the other failures. For example if their dataset is “Natural-Color Dataset”*, since the color of different vegetables and fruits can cause a prior bias for the model, then the user may want to mitigate failure modes that are only related to color. We will put the results for this dataset in our paper. Please see the spurious correlation outcomes related to color on CIFAR-100 as discussed in the previous response.
> Moreover, in “When does dough become a bagel? Analyzing the remaining mistakes on ImageNet” they have human-intervention to analyze the mistakes that models make on ImageNet, and they found that it can be because of multi-labeled examples like we did confirm in Figure 7. However, our main contribution is to selectively detect and improve failures in a way that is both interpretable and automatic. We also incorporated collective_debugging which is as of our knowledge the first work that improves the performance of a group of models with respect to only the failure set of one model with in each category, and demonstrated the well-known fact that models with same architectures have very similar features including what they learn and do not learn.

---

> ### Author Response · Authors · 2024-01-25
>
> > The effectiveness of the framework appears to be inconsistent ...
>
> We thank the reviewer for noticing this. To clarify the test accuracies, it may be helpful to go through the framework again. We divide all failures of a model into to categories seed and held-out in a way that they have nearly even numbers of data in each class of the dataset. We use the seed to generate more examples and then we test the final model on heldout. The number of data in seed dataset is much smaller comparing to the test set which has 20 examples per each class of ImageNet. There are also some cases that there is no failure for a class in the seed dataset (which we generated k examples for that failure). So it is more possible that for some classes in the test set we did not generate enough examples ro improve the test set accuracy as much as the heldout set which has the same number of examples per class as the seed set. This is a realistic scenario because the set that we use for debugging the model in small. However, to clarify this, we also show the clean and failure accuracies separately in the table below.
>
> | Models |  | Accuracies |  |
> | :---: | :---: | :---: | :---: |
> | Model category | model name | clean data | failures |
> |  | resnet18 | 0.9892 | $\mathbf{0 . 1 3 0 5}$ |
> |  | resnet34 | 0.9952 | $\mathbf{0 . 1 5 6 0}$ |
> | ResNet | resnet50 | 0.9931 | $\mathbf{0 . 1 3 9 4}$ |
> |  | resnet101 | 0.9991 | $\mathbf{0 . 1 6 9}$ |
> |  | resnet152 | 0.9943 | $\mathbf{0 . 1 5 2 5}$ |
>
> *Accuracy of ResNets on clean data and failures of CIFAR-100 after Individual Debugging. Our method does not harm the clean accuracy.
>
> > The experiments are insufficient, primarily ...
>
> Thank you for your suggestion. We hope that the new experiments on Cifar100 (color), and iNaturalist (background) and also the clean and failure accuracies will help us to prove our point with more experiments. We hope additional experiments with these two datasets are sufficient enough. Please let us know if we can strengthen up our results with any new experiments that you would suggest.
>
> > Enhance clarity by incorporating a figure ...
>
> This is a great suggestion. For your first request, do you think figure 3 provides this visual example? For example in the first image the detected background is sea, and after addition of a generated example of “A refrigerator in sea”, the model “ResNet18” will correctly classify this example as a refrigerator.
> The peuso-code will also be added to the paper for more clarity. Thanks for your idea
>
> > Clarify whether the observation of similar ...
>
> The observation has been identified in a previous paper which is cited in our paper. However, the main contribution of the paper is to use these detected similar failures to collectively improve performances of a group of model in a same category with the use of failures of only one model in this category, which is highlighted as the main contribution of the paper, and has not been investigated in prior works.

---

> > ### Comment · Reviewer_WmYW · 2024-02-10
> > **Thanks for the response but some concerns remain**
> >
> > I appreciate the authors for their detailed response and the additional experiments conducted on new datasets. While the response has partially addressed my initial concerns, two primary issues remain:
> >
> > [A] Similar to Reviewer nTBH, I believe that the submission should provide a detailed analysis of how the proposed method contributes to or affects the testing accuracy, in addition to the hold-out set accuracy.
> >
> > [B] Similarly, as noted by Reviewer ZC7d, the submission requires careful improvement in terms of writing and presentation.

---

> > > ### Author Response · Authors · 2024-02-12
> > >
> > > Thanks for reviewing our answers.
> > >
> > > > [A] Similar to Reviewer nTBH, I believe that the submission should provide
> > >
> > > As for your first question, we tested our framework on both clean and failure data separately, and we provided the results of this experiment in the answer to your initial questions. We have also added this result to our paper with further explanations.
> > >
> > >  > [B] Similarly, as noted by Reviewer ZC7d, the submission requires
> > >
> > > We revisited the paper, making efforts to address this aspect to the best of our observations.

---

### Review · Reviewer_VNVa · 2024-01-16

**Summary Of Contributions:**

The paper proposes to use a combination of ChatGPT and Stable Diffusion to guide the process of data augmentation (in order to create *harder* positives) for improving model performance against what is termed as "failure modes" in the paper, either on a single model or a group of them. The approach is devised in an end-to-end fashion to potentially automate the detection and treatment of such model failure modes.

**Audience:**

Yes

**Claims And Evidence:**

Yes

**Requested Changes:**

1-  Most requested changes are already covered in discussion around weaknesses. Beyond those:

The paper will benefit from another proof-read. Some examples:
- However, (w)e take a step further and delve into leveraging these consistencies to systematically mitigate shared failure modes
- For example, (A)ssociating objects with background  ...

2- What if chatGPT proposes uncommon backgrounds that have little to do with actual *hard* backgrounds in which the model fails? There is no binding between the debug set and uncommon backgrounds suggested by ChatGPT except for the `"target class", right? In this context, does "redundant" uncommon background mean repetitive recommendations only? Please address these with further elaborations in the paper.

3- The paper refers to a bunch of self-supervised learning (SSL) models, the likes of DINO, MoCo etc. How are the SSL models evaluated here - are they pretrained on the same bigger dataset first (say Imagenet) with no labels and the finetuned with limited samples? That can cause data leakage issues, this needs further clarification.

4 - Table 2, why doesn't test set performance improve much? those classes in the held out set exist there too, and the improved ("debugged") model should make an impact there as well, no? I do believe the experiments are not well explained. Please consider substantially improving the reporting of such results.

**Strengths And Weaknesses:**

**Strength**
- The paper has a clear narrative and is structured well.
- The idea is rather simple yet (seemingly) effective.
- Experimental results are interesting and somewhat promising.

**Weaknesses**
- The claim on being a comprehensive approach is not substantiated, neither through explanation nor experimentation. Nonetheless, I do not agree with that statement and think the main angle of novelty is being an autonomous approach. More on this in the following.
- Some choices of naming might be confusing. E.g. debugging: this might not resonate with the audience as what the paper does is more of an model enhancement than debugging. Another example is few-shot, this is a reserved word within this community and yet this only appears in the title and abstract, and that's it. Beyond that, no standard few-shot setting is explored in experimentation nor is any few-shot model used as baseline. **Requested change:** would be nice to stratify this as well as consider analyzing the impact on few-shot settings and models.
- In its related work study, I think the paper misses out on two areas of research with proximity and relevance.
    * Optimal adaptation of Diffusion models to generate more sensible samples for a downstream task (could be classification for instance) here is an example => https://openreview.net/pdf?id=WRpRPsU0VT,
   * More related, generating *hard positives and negatives* as augmentation for model enhancement. This is done using GANs and more recently with diffusion models.  See for instance SDEdit and Boomerangh and references therein: https://arxiv.org/pdf/2108.01073.pdf, https://openreview.net/forum?id=NYdThkjNW1.
  * Even closer to this work see GeNIe (https://arxiv.org/pdf/2312.02548.pdf) where hard examples are generated through *stable diffusion* for *few-shot* and *long-tail distribution* model performance improvement, two areas in which further experimentation can help substantiate the impact of this work as well. **Requested change:** consider improving the related work, and I do suggest extra experimentation to compare with standard augmentation approaches (your random debugging is good first step) such as CutMix and MixUp and the aforementioned references.
- Experimental setup is a bit confusing and experimentations need extensions and further elaboration. In my eyes, rather simple methodology here should be accompanied by more extensive experimentation to corroborate the impact of the work. More on this in requested changes.

---

> ### Author Response · Authors · 2024-01-25
>
> We thank the reviewer for their helpful comments on our paper.
>
> > The claim on being a comprehensive ...
>
> We thank the reviewer for their comment. The problem of spurious correlation has been studied for years, and it still has not yet been fully solved. Our approach has 3 main features. Firstly, it is automatic which is helpful when dealing with real-world datasets. Secondly, it is internpretable which is a really important manner in case of failure mode detection and analysis. Unlike many other recent work that lack the explainability for the detected failures (https://openaccess.thecvf.com/content/WACV2024/papers/Singla_Data-Centric_Debugging_Mitigating_Model_Failures_via_Targeted_Image_Retrieval_WACV_2024_paper.pdf). Thirdly, as of our best knowledge, this is the first work that perform collective_debugging (Improving the performance of a group of models by using only one model’s failures). Our experiments proved two important claims. Firstly, the problem of spurious correlation (e.g. background) is an important problem in DL models, since after applying the framework, we solve over 20% of all failure modes. Secondly, This problem can be solved only by adding targeted data without any architecture alteration.
> Moreover, we would really be happy if you have any suggestion on more experiments that can clarify our goal more. Besides, We added some more experiments to our paper that can be found below:
>
> | Models |  |  |  |  Accuracies |  |  |  |  |  |
> | :---: | :---: | :---: | :---: | :---: | :---: | :---: | :---: | :---: | :---: |
> | Model category | model name | Accuracy  before debugging |  | Accuracy of Individual Debugging (ours) |  |  | Accuracy of Random debugging |  |  |
> |  |  | Test | Test | seed | heldout |  | Test | seed | heldout |
> |  | resnet18 | 0.7551 | 0.7789 | 0.3323 | $\mathbf{0 . 2 8 1 8}$ |  |  0.7591 | 0.1901  |  0.1878 |
> |  | resnet34 | 0.7666 | 0.7994 | 0.3586 | $\mathbf{0 . 2 9 2 3}$ |  | 0.7715 | 0.1902  |  0.1947 |
> | ResNets | resnet50 | 0.7730 | 0.7993 | 0.3614 | $\mathbf{0 . 3 0 0 8}$ |  | 0.7715  |  0.1971 | 0.1858 |
> |  | ResNet101 | 0.7748 | 0.8123 | 0.3877 | $\mathbf{0 . 3 2 4 9}$ |  | 0.7811 | 0.2001  |  0.1930 |
> |  | resnet152 | 0.7769 | 0.8065 | 0.3842 | $\mathbf{0 . 3 1 8 4}$ |  | 0.7797 | 0.2139  |  0.2152 |
>
> *Accuracy of our method compared to the Random_debugging on CIFAR-100 considering color spurious correlations. Note that the accuracy of models on debug_seed and debug_heldout was zero before debugging. After applying our debugging method, we gain above ∼ 28% improvements in accuracies for all models, showcasing that more than ∼ 28% of model errors in the heldout set come from wrong background associations.
>
> *The prompt used in the above table to input ChatGPT is "What is an uncommon color that <class_name> may possess?", and the prompt for generating more data with Stable diffusion is "A photo of a <color> <class_name>"
>
> | Models |  | Accuracies |  |
> | :---: | :---: | :---: | :---: |
> | Model category | model name | clean data | failures |
> |  | resnet18 | 0.9892 | $\mathbf{0 . 1 3 0 5}$ |
> |  | resnet34 | 0.9952 | $\mathbf{0 . 1 5 6 0}$ |
> | ResNet | resnet50 | 0.9931 | $\mathbf{0 . 1 3 9 4}$ |
> |  | resnet101 | 0.9991 | $\mathbf{0 . 1 6 9}$ |
> |  | resnet152 | 0.9943 | $\mathbf{0 . 1 5 2 5}$ |
>
> *Accuracy of ResNets on clean data and failures of CIFAR-100 after Individual Debugging. Our method does not harm the clean accuracy.
>
> | Models |  |  |  |  Accuracies |  |  |  |  |  |
> | :---: | :---: | :---: | :---: | :---: | :---: | :---: | :---: | :---: | :---: |
> | Model category | model name | Accuracy before debugging |  | Accuracy of Individual Debugging (ours) |  |  | Accuracy of Random debugging |  |  |  |
> |  |  | Test | Test | seed | heldout | Test | seed | heldout |  |
> |  | resnet50 | 0.7581 | 0.7852 | 0.3612 | $\mathbf{0 . 3 0 8 2}$ | 0.7614 | 0.1348 | 0.1352 |  |
> | ResNet | resnet101 | 0.7863 | 0.8148 | 0.3910 | $\mathbf{0 . 3 1 7 3}$ | 0.7899 | 0.1377 | 0.1384 |  |
> |  | resnet152 | 0.7996 | 0.8252 | 0.3916 | $\mathbf{0 . 3 1 9 4}$ | 0.8015 | 0.1485 | 0.1463 |  |
>
> *Accuracy of our method compared to the Random_debugging on iNaturalist-2018 considering
> background spurious correlations. Note that the accuracy of models on debug_seed and debug_heldout was
> zero before debugging.
>
> > Some choices of naming might be ...
>
> Thanks for your valuable insight. In figure 5, b, we showed the effect of different Ks, which is the number of generated data per class. However, we would be more than happy to run more experiments if you have anything in mind.
> Moreover, We would consider changing the name “debugging” to something that can better clarify the purpose of the paper. Perhaps “refinement” would be a better choice?

---

> ### Author Response · Authors · 2024-01-25
>
> > In its related work study, I think the paper misses ...
>
> Thanks you for this helpful suggestion. We are working on the new experiments to compare our method to other existing methods. We will share the results as soon as they are done. We are also changing the related work section a bit to cover the two areas you mentioned in your comment.
>
> > Experimental setup is a bit confusing and experimentations need ..
>
> Thanks for your comment. We added more experiments and you can find them in the answer to your first comment.
>
> > The paper will benefit from another proof-read. Some examples: ...
>
> Thanks for mentioning this. We corrected these problems in our paper along with other mistakes.
>
> > What if chatGPT proposes uncommon ...
>
> We will explain this more in the paper. It is correct that ChatGPT recommendations may be not aligned with the real backgrounds in the datasets, because it does not have knowledge about the actual data in the dataset. It will only give an approximation of what can the rare backgrounds can be. However, In most cases, it will suggest backgrounds that are close to what failures occur in. In future work, we would like to explore this more to propose a method that can more precisely suggest uncommon settings. We will add this explanation in the discussion part of the paper.
> It is correct. “Redundant” means repetitive recommendations.
>
> > The paper refers to a bunch of self-supervised learning ...
>
> They are pretrained on the training set of ImageNet, which we did not use for evaluation and model refinement (debugging).
>
> > Table 2, why doesn't test set performance ...
>
> We thank the reviewer for noticing this. To clarify the test accuracies, it may be helpful to go through the framework again. We divide all failures of a model into to categories seed and held-out in a way that they have nearly even numbers of data in each class of the dataset. We use the seed to generate more examples and then we test the final model on heldout. The number of data in seed dataset is much smaller comparing to the test set which has 20 examples per each class of ImageNet. There are also some cases that there is no failure for a class in the seed dataset (which we generated k examples for that failure). So it is more possible that for some classes in the test set we did not generate enough examples ro improve the test set accuracy as much as the heldout set which has the same number of examples per class as the seed set. This is a realistic scenario because the set that we use for debugging the model in small. However, to clarify this, we also show the clean and failure accuracies separately in the table below.
>
> | Models |  | Accuracies |  |
> | :---: | :---: | :---: | :---: |
> | Model category | model name | clean data | failures |
> |  | resnet18 | 0.9892 | $\mathbf{0 . 1 3 0 5}$ |
> |  | resnet34 | 0.9952 | $\mathbf{0 . 1 5 6 0}$ |
> | ResNet | resnet50 | 0.9931 | $\mathbf{0 . 1 3 9 4}$ |
> |  | resnet101 | 0.9991 | $\mathbf{0 . 1 6 9}$ |
> |  | resnet152 | 0.9943 | $\mathbf{0 . 1 5 2 5}$ |
>
> *Accuracy of ResNets on clean data and failures of CIFAR-100 after Individual Debugging. Our method does not harm the clean accuracy.

---

> ### Comment · Reviewer_VNVa · 2024-01-26
> **Good addition**
>
> Thank you for the extra experiments. Please make sure all tables and pivotal results and takeaways are much better explained. Yes I think refinement or enhancement might be a better choice.
>
> *Note*: since you didn't pick different titles for your response, my reactions to your responses will not have a distinct title. So you need to track based on connect and make the connection.

---

> > ### Author Response · Authors · 2024-01-30
> >
> > Thank you for reviewing our responses addressing your concerns. We trust that the majority of issues have been resolved. We plan to submit the revised version of the paper, incorporating all your valuable comments, at the earliest opportunity.

---

> > > ### Comment · Reviewer_VNVa · 2024-02-04
> > > **Comments not Addressed**
> > >
> > > Thanks for you efforts so far. Please note that a bunch of my points are not addressed yet (such as comparing against other rather straightfoward but extremely relevant baselines, better related work, improving the explanation provided for tables etc, for which  I am still waiting for the revised version of the draft. Note that soon we need to make a final decision.

---

> > > > ### Author Response · Authors · 2024-02-12
> > > >
> > > > We thank the reviewer for their careful evaluation of our paper. We revised the paper taking all the comments into account and we have submitted it. Please let us know if there is still a place for improvement.

---

> > > > > ### Comment · Reviewer_VNVa · 2024-02-14
> > > > > **Last remarks**
> > > > >
> > > > > My last two cents:
> > > > > - I still believe ChatGPT has no guarantee to propose meaningful hard backgrounds. This is a pivotal assumption, and the margin we observe using ChatGPT might fully wear out in larger datasets. In my eyes, one might have to resort to data mining to extract such backgrounds. Nonetheless, thanks for your explanations.
> > > > > - I believe there is a difference between random augmentation techniques and guided ones for generating hard positives and negatives. As such, I suggested to compare at least against some more advanced augmentation techniques to substantiate the impact of you proposed approach. That is a rather straightforward exercise, and those built around Diffusion models could easily be assessed in your current pipeline.

---

> ### Comment · Reviewer_VNVa · 2024-01-26
> **Looking forward to the results and improved related work**
>
> - Thanks, looking forward to seeing the comparative results and the improved related work.
>
> - "*It is correct that ChatGPT recommendations may be not aligned with the real backgrounds in the datasets, because it does not have knowledge about the actual data in the dataset. ... However, In most cases, it will suggest backgrounds that are close to what failures occur in. In future work ...*" I'm not convinced here. The process is not controlled in any shape or form, how can that lead to suggestion of relevant background in most cases. This is a pivotal step for which no guarantees are in place. What is the impact?
>
> - ''They are pretrained on the training set of ImageNet, which we did not use for evaluation and model refinement (debugging).''. Thanks for the clarification. But do they see data from any of those classes of failure mode in the test set? or the classes are mutually exclusive from the ones the SSL methods are trained on?

---

> > ### Author Response · Authors · 2024-01-30
> >
> > We express our gratitude to the reviewer for revisiting our responses.
> >
> > > "It is correct that ChatGPT recommendations may be not aligned with the real backgrounds in the datasets, because it does not have knowledge about the actual data in the dataset. ... However, In most cases, ...
> >
> > Regarding the suggestion about ChatGPT providing backgrounds closely aligned with failure occurrences, we acknowledge the reviewer's concern regarding the absence of absolute guarantees for ChatGPT-generated backgrounds. Having such guarantees would imply a comprehensive understanding of all failure cases related to spurious correlations in a given dataset, rendering the need for detection obsolete. However, this scenario is not reflective of real-world conditions. To automatically generate failure modes, deploying a large language model is essential. We opted for ChatGPT, a widely used choice in similar contexts (as evidenced by, for instance, https://arxiv.org/pdf/2310.00164.pdf).
> >
> > The efficacy of employing ChatGPT is demonstrated in Table 2, where we showcase significant improvements by leveraging the spurious correlations suggested by ChatGPT along with the captioning capabilities of CLIP. This outperforms scenarios where we solely rely on class names, highlighting the utility of ChatGPT in enhancing our framework and avoiding human-intensive processes.
> >
> > Furthermore, we evaluated our framework with a set of predefined spurious associations based on our dataset knowledge. The results are presented in the table below:
> >
> > | Models |  | Accuracies |  |
> > | :---: | :---: | :---: | :---: |
> > | Model category | Model name |  |  Accuracy of Individual Debugging|
> > |  |  |  seed | heldout |
> > |  | resnet18 | 0.2260 | $\mathbf{0 . 1 6 9 0}$ |
> > |  | resnet26 | 0.2295 | $\mathbf{0 . 1 7 5 1}$ |
> > | ResNets | resnet34 | 0.2294 | $\mathbf{0 . 1 7 8 8}$ |
> > |  | resnet50 | 0.2354 | $\mathbf{0 . 1 8 0 6}$ |
> > |  | resnet101 | 0.2384 | $\mathbf{0 . 1 8 4 7}$ |
> > |  | resnet152 | 0.2402 | $\mathbf{0 . 1 8 5 7}$ |
> > |  | ViTs8 | 0.2310 | $\mathbf{0 . 2 0 7 3}$ |
> > |  | ViTs16 | 0.2231 | $\mathbf{0 . 2 0 4 4}$ |
> > | DINO | ViTb8 | 0.2492 | $\mathbf{0 . 2 2 7 1}$ |
> > |  | ViTb16 | 0.2359 | $\mathbf{0 . 2 1 1 8}$ |
> > |  | resnet50 | 0.1904 | $\mathbf{0 . 1 7 4 2}$ |
> >
> > The above table shows the result for when we choose a set of 40 predefined backgrounds with respect to the knowledge we have from the ImageNet dataset. These backgrounds are ["in a blur background", "on a leaf", "in water", "on soil", "on a plate", "in a hand", "on a sofa", "in garden", "in jungle", "in cave", "on snow", "in a plane background", "in yard", "outdoor", "in shore", "in the sky", "indoor", "on a wall", "on a tree", "on a table", "in a street", "on a rock", "in a airplane", "in cage", "with sun", "in a mountain", "in metro", "on grass", "in shelf", "on rails", "with a person", "on bed", "in a playground", "in a kitchen", "on floor", "on glass", "on wood", "at a party", "in a wardrobe", "in a restaurant", "in a bucket"].
> >
> > The improvement we observe from using ChatGPT is more than that of a predefined set which is suggested by a human. This is understandable since Large Language Models like ChatGPT have been trained on massive amounts of data and they have more knowledge regarding this manner. This along with the automatic generation of spurious correlations is the reason we preferred ChatGPT, with all its shortcomings, over relying on humans.
> >
> > We hope this explanation clears the need for using ChatGPT. Moreover, we would really appreciate if you have any more suggestion regarding this manner.
> >
> >
> > > ''They are pretrained on the training set of ImageNet, which we did not use for evaluation and model refinement (debugging).''. Thanks for the clarification. But do they ...
> >
> > In our approach, we utilize models pretrained on ImageNet. Subsequently, we partition the validation set into two distinct subsets: refinement and test sets. The refinement set comprises 30 images per class, while the test set consists of 20 images per class. It's noteworthy that the classes remain consistent across the training, refinement, and test phases. However, the specific data within these classes varies between each respective phase. This design allows us to investigate the potential efficacy of augmenting data for classes with limited representation under diverse spurious conditions, aiming to address associated challenges.

---

> > ### Author Response · Authors · 2024-02-12
> >
> > > It is correct that ChatGPT recommendations may be not aligned with the real backgrounds in the datasets, because it does not have knowledge about the actual data in the dataset. ... However, In most cases, it will suggest backgrounds that are close to what failures occur in. In future work ..." I'm not convinced here. ...
> >
> > We acknowledge the reviewer's feedback. The challenge we face lies in the limited options for addressing this issue. Existing literature primarily proposes two methods: 1) employing a predetermined set of samples derived from a specified spurious correlation (in this case, the background), and 2) leveraging a large language model like ChatGPT to automatically propose such backgrounds. In our submission, we have presented results for the first approach, acknowledging its reliance on prior dataset knowledge and its comparatively lower accuracy improvement compared to ChatGPT. While we recognize that we cannot exert direct control over ChatGPT's suggestions, it currently represents our most effective strategy. We are actively exploring alternative approaches for future research to further enhance our methodology.
> >
> > > ''They are pretrained on the training set of ImageNet, which we did not use for evaluation and model refinement (debugging).''. Thanks for the clarification. But do they see data from any of those classes of failure mode in the test set? or the classes are mutually exclusive from the ones the SSL methods are trained on?
> >
> > The categories assigned to the classes remain consistent across all stages, encompassing training, debugging, and testing phases. In the context of ImageNet, the 1000 classes persist throughout. Our data partitioning ensures that instances are distinctly allocated for each step, maintaining mutual exclusivity.

---

### Review · Reviewer_2AQA · 2024-01-19

**Summary Of Contributions:**

Proposes a technique that leverages off-the-shelf foundation models (CLIP, ChatGPT, Stable Diffusion) to identify and mitigate model failures arising from spurious context correlations. The technique is shown to lead to large performance improvements across a range of models pretrained on standard object recognition datasets.

**Audience:**

Yes

**Claims And Evidence:**

Yes

**Requested Changes:**

Please see weaknesses above. In particular, I would like to see additional experiments for weaknesses  D-E and clarifications for A-C.

**Strengths And Weaknesses:**

Strengths

– The paper is clear and easy to follow
– The main findings of the paper are validated across a wide range of model backbones

Weaknesses

A) A few recent works [A, B] have proposed similar approaches of generating images of rare subpopulations with language guidance. The paper specializes this approach to deal with the application of identifying spurious context correlations (e.g. by only asking for rare contexts), but it's technical contributions overall are limited.

B) The paper claims that its approach can be generalized easily to other kinds of spurious correlations beyond background, but does not provide details or evidence of the same.


C) The absolute performance obtained after mitigation seem low (e.g ~20-30% in Table 2 on seed set). Is this a function of the small debug_train set, the few-shot retraining strategy, or something else?

D) The approach appears to assume that all failures are due to spurious context correlations (since that is how the debug set is collected), but does not consider or control for other failure modes (eg. label noise, atypicality, etc.). More specifically, the failure modes are textualized using CLIP similarity to "A photo of <class> in <uncommon_background>", but the paper lacks an evaluation of the effectiveness of this strategy.

E) While the collective failure inspection idea is interesting, the paper would be strengthened by visualizations + evaluations of the accuracy of the identified failure modes (perhaps, by studying if the proposed approach can discover a known spurious correlation in an existing diagnostic dataset, or even better rank backgrounds based on their impact on performance.

[A] Dunlap et al., Diversify Your Vision Datasets with Automatic Diffusion-Based Augmentation, NeurIPS 2023
[B] Kattakinda et al., Invariant Learning via Diffusion Dreamed Distribution Shifts, arXiv 2022

---

> ### Author Response · Authors · 2024-01-25
>
> > A few recent works [A, B] ...
>
> Thanks for mentioning this important point. It is quite clear that the failure modes of a model on a dataset can be due to many different reasons. There are many works (cited in the related work section) that try to improve the performance of a model on a dataset without considering any specific failure case (for example background, color association). Although in most cases, they successfully improve the performance, one big problem is that most of the proposed method is not interpretable. In this paper, we focused on the interpretability of the framework as well to increase the knowledge about failures that a model makes. This will give us insight on how to refine the model with respect to different failure cases, and it makes it possible to look into the reason of the failure. In addition, the detection of failure modes with respect to one aspect at a time has another benefit which is the user can select the most important aspect to refine based on their dataset. For example if their dataset is “Natural-Color Dataset”*, since the color of different vegetables and fruits can cause a prior bias for the model, then the user may want to mitigate failure modes that are related to color. We will put the results for this dataset in our paper.
>
> | Models |  |  |  |  Accuracies |  |  |  |  |  |
> | :---: | :---: | :---: | :---: | :---: | :---: | :---: | :---: | :---: | :---: |
> | Model category | model name | Accuracy  before debugging |  | Accuracy of Individual Debugging (ours) |  |  | Accuracy of Random debugging |  |  |
> |  |  | Test | Test | seed | heldout |  | Test | seed | heldout |
> |  | resnet18 | 0.7551 | 0.7789 | 0.3323 | $\mathbf{0 . 2 8 1 8}$ |  |  0.7591 | 0.1901  |  0.1878 |
> |  | resnet34 | 0.7666 | 0.7994 | 0.3586 | $\mathbf{0 . 2 9 2 3}$ |  | 0.7715 | 0.1902  |  0.1947 |
> | ResNets | resnet50 | 0.7730 | 0.7993 | 0.3614 | $\mathbf{0 . 3 0 0 8}$ |  | 0.7715  |  0.1971 | 0.1858 |
> |  | ResNet101 | 0.7748 | 0.8123 | 0.3877 | $\mathbf{0 . 3 2 4 9}$ |  | 0.7811 | 0.2001  |  0.1930 |
> |  | resnet152 | 0.7769 | 0.8065 | 0.3842 | $\mathbf{0 . 3 1 8 4}$ |  | 0.7797 | 0.2139  |  0.2152 |
>
> *Accuracy of our method compared to the Random_debugging on CIFAR-100 considering color spurious correlations. Note that the accuracy of models on debug_seed and debug_heldout was zero before debugging. After applying our debugging method, we gain above ∼ 28% improvements in accuracies for all models, showcasing that more than ∼ 28% of model errors in the heldout set come from wrong background associations.
>
> *The prompt used in the above table to input ChatGPT is "What is an uncommon color that <class_name> may possess?", and the prompt for generating more data with Stable diffusion is "A photo of a <color> <class_name>.".
>
> Moreover, the works A, and B generate diverse data given a dataset. Our method has several advantages. Firstly, it generates additional data only for failure cases which is both time and cost efficient. Secondly, it is more specific (A, and B generates additional data for all the instances of the given dataset which may not be necessary for the case of failure mitigation.). Another contribution of our paper which is as of our knowledge the first result in this context, is collective debugging which is refinement of a group of model by having the failures of only one model of the group. This is even more efficient in both time and cost.
>
> > The paper claims that its approach can be ...
>
> Thanks for bringing this up. We are adding another example of spurious correlation (Color) to our results to more accurately show our claim. The results for this can be found in the answer to your previous comment.
>
> > The absolute performance obtained after mitigation ...
>
> Thanks for asking this. The reason is that the failures of a model on a dataset can be due to many different reasons as you mentioned in your first question. This result shows that only about 20%-30% of failures were due to the spurious background associations.
>
> > The approach appears to assume that all failures ...
>
> We thank the reviewer for asking this question. As we explained in the answer to your first question. It is more applicable for many different usages if we can specify the kind of failure mode we want to mitigate. For example if their dataset is “Natural-Color Dataset”*, since the color of different vegetables and fruits can cause a prior bias for the model, then the user may want to mitigate failure modes that are related to color. Our specific focus in this paper is spurious correlations, and to show the effectiveness of how much mitigating a selected spurious correlation can effect the model performance and making sure that the gain is due to the background association, we also performed random_debugging in which we only use class_names to generate additional data. Table 2 shows that taking the spurious correlation (here background) into account definitely resolves more failures than random_debugging.

---

> ### Author Response · Authors · 2024-01-25
>
> > While the collective failure ...
>
> Thanks for suggesting this. We found some examples to illustrate this more visually. You can find the visualization in the link below:
> https://ibb.co/VSPBCvd
> The numbers in the image show the percentage of the shared failures that are related to the mentioned background. In the figure, we showed the most three common failure backgrounds.

---

> > ### Comment · Reviewer_2AQA · 2024-02-01
> > **Thank you for the detailed response**
> >
> > Thank you for the detailed response. I appreciate the additional experiments, but have two lingering concerns:
> >
> > 1) In my opinion, requiring prior knowledge of the potential source of bias (color / background) reduces the utility and versatility of the method as it can be both expensive and challenging to do so for real-world datasets.
> >
> > 2) While I appreciate the clarification that the 0$\to$28% accuracy bump is likely due to fixing spurious correlations alone, without an oracle it is challenging to qualify as good – eg. Is it not possible that maybe perfectly “fixing” spurious correlations would increase performance to 50%, but this method only gets you to 28%? Perhaps a toy dataset like Waterbirds / CelebA / NICO++ etc. known to have a high degree of spurious correlation would be more convincing.
> >
> > I appreciate the additional qualitative results but found the description lacking. I would encourage the authors to revise the paper with a more detailed discussion.

---

> > > ### Author Response · Authors · 2024-02-12
> > >
> > > We thank the reviewer for carefully reading our comment.
> > >
> > > > In my opinion, requiring prior knowledge of the potential source of bias (color / background) ...
> > >
> > > Our literature review process for writing this paper has shown that many works have focused on background spurious correlations such as (https://arxiv.org/pdf/2211.10370.pdf, https://arxiv.org/pdf/2110.03804.pdf, https://arxiv.org/pdf/2006.09994.pdf). Our approach was first developed to only address background correlations, however upon more experiments, we have found the applicability of our approach on other types of correlations.
> > >
> > > > While I appreciate the clarification that the ...
> > >
> > > Thanks for your helpful comment. We agree that perfectly mitigating this issue may improve the final results further. However, this may need human-intervention to carefully inspect data and find all the possible spurious correlations. In this paper, we are trying to achieve this but also be automatic, interpretable, and memory, and time efficient.
> > > Although there are no perfect way of finding all spurious correlations and mitigate them, we added a section in the appendix in which we included the results of some baselines to better compare our framework's improvement with other existing methods. Considering these baselines and also the results for random_refinement, which clearly shows the importance of taking background correlations into account, our framework seems to find more spurious correlations and mitigate them comparing to other baselines, with the advantage of adding a very few additional data to the training set (3 per class) which is far fewer than other existing works (For example, in https://openaccess.thecvf.com/content/WACV2024/papers/Singla_Data-Centric_Debugging_Mitigating_Model_Failures_via_Targeted_Image_Retrieval_WACV_2024_paper.pdf they add 30,000 data per class which is very expensive in terms of time and memory).
> > > Moreover, In the revised version of the paper, we included some more examples on another dataset iNaturalist-2018 (background) and CIFAR-100 (color).
> > >
> > >
> > > Please kindly read the revised version that we submitted and let us know if there is still any part that we can further improve.

---

### Review · Reviewer_nTBH · 2024-01-20

**Summary Of Contributions:**

This work proposes a general approach to mitigating image classification failures caused by spurious foreground-background correlations by utilizing external tools including ChatGPT, CLIP, and Stable Diffusion. More specifically, 1) ChatGPT is first prompted to propose a set of less likely backgrounds for a given class, then 2) CLIP zero-shot classification is used to attribute the failure cases to the proposed backgrounds, and finally 3) Stable Diffusion synthesizes images depicting such objects in unusual backgrounds causing failures. Experiments on CIFAR and ImageNet demonstrate that these additional training data can 1) reduce errors on validation images that are unseen during the debugging process, and 2) can generalize to classification models with similar architectures.

**Audience:**

Yes

**Claims And Evidence:**

Yes

**Requested Changes:**

Most of the requested changes have been discussed in the weakness section above. In addition to these requested changes, there are some minor suggestions:

- In Figure 2, the axes around each image are unnecessary and could be removed.

- There are some typos to be fixed. Examples include:
    - Section 4.3: “Clip” -> “CLIP,” “figure 2” -> “Figure 2.”
    - Section 4.4: Please keep notations like “debug_train” and “$\lambda$” consistent.

**Strengths And Weaknesses:**

Strengths
1. The method is simple yet effective, and can benefit from the very recent advances in large language models and vision language models.

2. The collective failure mitigation shows generalization within a family of models, which is computation and data efficient.

3. This work shows that resolving spurious foreground-background correlations is a promising approach to enhancing recognition tasks.

Weaknesses
1. [Test Accuracy] Further analysis on the test set accuracy is needed.

    - Although the proposed method leads to significantly improved accuracy on a validation set “debug_heldout” that is unseen during the debugging process, the test set accuracy is only marginally improved (in most cases <+1%, as shown in Tables 2 and 3).

    - Moreover, when increasing the hyper-parameter $\lambda$, the test accuracy and debug_heldout accuracy show different trends. Intuitively, the validation set and test set follow the same data distribution.

    [Requested Change] There should be some discussion regarding the different effects of the proposed method on the two sets.

2. [Decomposition of Errors] There are multiple failure modes in image classification including noisy labels and rare backgrounds (Figure 7). The method proposed in this work is only designed for mitigating the failures caused by backgrounds. It is not very clear whether the proposed method can reduce other types of errors (e.g., via improved classification) as well, and whether the proposed method can greatly improve the overall accuracy. Again, from the test accuracy, it seems that the improvement is not very significant.

    [Requested Change] The authors may consider decomposing a subset of the validation or test errors and show the improvement with respect to each type or errors.

3. [Coverage of Failures] This method relies on ChatGPT to automatically propose a set of unfamiliar foreground-background pairs. The proposals are finite and may be arbitrary, and consequently, may not be able to cover all failure cases in the debug set. Especially, for a given foreground object class, ChatGPT may tend to propose backgrounds that are very unlikely or even impossible to appear in a natural image. In such cases, it is unclear how to ensure the coverage of failure modes in the debug set.

4. [Feature Extraction] This work only uses the synthesized images to retrain the linear classification head, but the features extracted from previous layers are kept unchanged.

    [Requested Change ]The authors may consider slightly tuning the feature extractor as well and check whether it can improve the classification performance.

---

> ### Author Response · Authors · 2024-01-25
>
> We thank the reviewer for their valuable insights.
>
> > Although the proposed method leads ...
>
> We thank the reviewer for noticing this. To clarify the test accuracies, it may be helpful to go through the framework again. We divide all failures of a model into to categories seed and held-out in a way that they have nearly even numbers of data in each class of the dataset. We use the seed to generate more examples and then we test the final model on heldout. The number of data in seed dataset is much smaller comparing to the test set which has 20 examples per each class of ImageNet. There are also some cases that there is no failure for a class in the seed dataset. So it is more possible that for some classes in the test set we did not generate enough examples ro improve the test set accuracy as much as the heldout set which has the same number of examples per class as the seed set. This is a realistic scenario because the set that we use for debugging the model in small. However, to clarify this, we also show the clean and failure accuracies separately in the table below.
>
> | Models |  |  |  |  Accuracies |  |  |  |  |  |
> | :---: | :---: | :---: | :---: | :---: | :---: | :---: | :---: | :---: | :---: |
> | Model category | model name | Accuracy  before debugging |  | Accuracy of Individual Debugging (ours) |  |  | Accuracy of Random debugging |  |  |
> |  |  | Test | Test | seed | heldout |  | Test | seed | heldout |
> |  | resnet18 | 0.7551 | 0.7789 | 0.3323 | $\mathbf{0 . 2 8 1 8}$ |  |  0.7591 | 0.1901  |  0.1878 |
> |  | resnet34 | 0.7666 | 0.7994 | 0.3586 | $\mathbf{0 . 2 9 2 3}$ |  | 0.7715 | 0.1902  |  0.1947 |
> | ResNets | resnet50 | 0.7730 | 0.7993 | 0.3614 | $\mathbf{0 . 3 0 0 8}$ |  | 0.7715  |  0.1971 | 0.1858 |
> |  | ResNet101 | 0.7748 | 0.8123 | 0.3877 | $\mathbf{0 . 3 2 4 9}$ |  | 0.7811 | 0.2001  |  0.1930 |
> |  | resnet152 | 0.7769 | 0.8065 | 0.3842 | $\mathbf{0 . 3 1 8 4}$ |  | 0.7797 | 0.2139  |  0.2152 |
>
> *Accuracy of our method compared to the Random_debugging on CIFAR-100 considering color spurious correlations. Note that the accuracy of models on debug_seed and debug_heldout was zero before debugging. After applying our debugging method, we gain above ∼ 28% improvements in accuracies for all models, showcasing that more than ∼ 28% of model errors in the heldout set come from wrong background associations.
>
> *The prompt used in the above table to input ChatGPT is "What is an uncommon color that <class_name> may possess?", and the prompt for generating more data with Stable diffusion is "A photo of a <color> <class_name>.".
>
> > Moreover, when increasing ...
>
> This is a great point. The reason for this is that if we increase the lambda substantially, the clean examples in the test set may be classified incorrectly since the generated data and the initial training data are from two different distributions. However, the debug_heldout set has no clean example by design. Therefore, no clean example will be incorrectly classified after the debugging process. The initial training data were not good enough for the model to learn the heldout set and when we increase lambda the model will pay more attention to generated data which are more informative for the model to gain good results on heldout set. Therefore, although the test set and heldout set have are from the same distribution, we should not increase the lambda to resolve more failures in heldout since it may cause the clean examples in the test set to be incorrectly classified after. This trade-off is illustrated in figure 5, b.
>
> > [Decomposition of Errors] There are multiple failure  ...
>
> Thanks for your question. The focus in our paper is on spurious correlation failures. Figure 7 was intended to show different types of failure modes that are very common in deep learning. Prior work have investigated all these 3 types of failures. However, for spurious correlation, they mostly work on background association, but our framework was designed to be general to other types of spurious correlation. Please refer to the color spurious correlation mitigation experiment on Cifar-100 in the answer to your first comment.

---

> ### Author Response · Authors · 2024-01-26
>
> > [Coverage of Failures] This method ...
>
> We will explain this more in the paper. It is correct that ChatGPT recommendations may be not aligned with the real backgrounds in the datasets, because it does not have knowledge about the actual data in the dataset. It will only give an approximation of what can the rare backgrounds can be. However, In most cases, it will suggest backgrounds that are close to what failures occur in. Moreover, we collect uncommon backgrounds suggested by ChatGPT for each class of data individually, but for choosing the background with CLIP, we combine all these uncommon backgrounds and perform the zero-shot classification task. Therefore, it will cover more real backgrounds at the end. In future work, we would like to explore this more to propose a method that can more precisely suggest uncommon settings. We will add this explanation in the discussion part of the paper.
>
> > [Feature Extraction] This work only uses the ...
>
> This is a valuable suggestion. Our goal was to make the minimum amount of change to the model, so that a user would not have to download the whole model again, and only a linear head retraining will be enough for the downstream task. Moreover, The framework tend to be general to all kinds of spurious correlations. Finetuning will need human observation to choose the related layers of the model which are responsible for the specific spurious correlation. For example, for color it has been investigated in prior work that ResNets and ViTs learn it in their very first layers, which is not the case for backgrounds.
>
> > In Figure 2, the axes ...
>
> Thank you for your attention. We would definitely apply these changes to our paper.

---

> > ### Comment · Reviewer_nTBH · 2024-02-05
> > **Remaining Concerns**
> >
> > Thank you very much for the detailed response. It is great to incorporate the discussion here into the paper and enhance this work. However, some of the previous concerns still remain:
> >
> > - [Test Accuracy] Indeed, the proposed method can bring up the accuracy of a specific subset from zero to about 30%, but we do not see a larger *generalizable* gain on the test set (also observed by Reviewer WmYW). It could be the case that some new data with rare foreground-background combinations are synthesized and the model is fitted to avoid errors on such type of data, but at the same time, other types of errors arise. After all, the generated data and the initial training data (and the test data) follow quite different distributions. Also, as suggested by Reviewer VNVa, it might be helpful to compare with some simple and solid baselines (e.g., standard data augmentation) that also reduce errors.
> >
> > - [Coverage of Failures] Since ChatGPT does not directly see the mis-classified image, the current method may not be able to correctly identify the background and leads to unsuccessful generalization (also mentioned by Reviewer ZC7d). The authors reply that "in most cases" ChatGPT can suggest backgrounds close to the actual failure case, but it is still unclear how to measure this accuracy and analyze the effectiveness of background identification.

---

> > > ### Author Response · Authors · 2024-02-12
> > >
> > > > Thank you very much for the detailed response. It is great to incorporate the discussion here into the paper and enhance this work ...
> > >
> > > We thank the reviewer for carefully reading our comments, and for their helpful comments that helped us to improve our paper. Please kindly review the revised version of our paper which we submitted.
> > >
> > >  > [Test Accuracy] Indeed, the proposed method can bring up the accuracy of a specific subset from zero to about 30%...
> > >
> > > We performed some new experiments to show the accuracy of the retrained model on clean data (the ones that were correctly classified before the retraining) and the failures individually. We put this result on the revised version of our paper. These results show that the retrained model does not decrease the accuracy of the clean data (1-2%) which is negligible comparing to the gain we get on failures.
> > > We also compared our method with some baselines in the appendix.
> > >
> > > > [Coverage of Failures] Since ChatGPT does not directly see the mis-classified image ...
> > >
> > >
> > > We acknowledge the reviewer's feedback. The challenge we face lies in the limited options for addressing this issue. Existing literature primarily proposes two methods: 1) employing a predetermined set of samples derived from a specified spurious correlation (in this case, the background), and 2) leveraging a large language model like ChatGPT to automatically propose such backgrounds. In our submission, we have presented results for the first approach, acknowledging its reliance on prior dataset knowledge and its comparatively lower accuracy improvement compared to ChatGPT. While we recognize that we cannot exert direct control over ChatGPT's suggestions, it currently represents our most effective strategy. We are actively exploring alternative approaches for future research to further enhance our methodology.

---

### Review · Reviewer_ZC7d · 2024-01-28

**Summary Of Contributions:**

The paper proposes a method to identify and mitigate failure modes using multiple foundation models. To identify failure modes, the method uses ChatGPT to enumerate possible uncommon backgrounds of a certain class. Then, CLIP is used to pseudo-label uncommon backgrounds for misclassified images. To mitigate failure modes, the method uses Stable Diffusion to generate images using prompts of “{class_name} in {uncommon_background},” which are used to retrain the linear head. The experiments show that (1) the proposed method can identify failure modes (Figure 3), (2) similar models (e.g., different sizes of ResNet) share similar failure modes (Figure 4), (3) failure modes can be mitigated in both individual (Sec. 5.3.1) and collective manners (Sec. 5.3.2).

**Audience:**

Yes

**Broader Impact Concerns:**

No ethical concerns from my perspective.

**Claims And Evidence:**

No

**Requested Changes:**

I request the authors make the following changes based on the weaknesses of the paper:

1. Evaluate the generalization to background categories not listed by ChatGPT.
2. Discuss the generalization capability of unknown shortcut types.
3. Clarify the problem where misclassified images have backgrounds not listed by ChatGPT.
4. Discuss the problem of only using misclassified images.
5. Discuss the failure modes of CLIP.
6. Clarify how debug seed and debug heldout sets are constructed.
7. Clarify why retraining the linear head. Shoe results of end-to-end retraining if possible.
8. Discuss the problem of the failure modes of Stable Diffusion.
9. Evaluate the method on existing benchmarks.
10. Discuss, cite, and compare with existing failure identification and mitigation methods
11. Add the results of CLIP as mentioned in the paper.
12. Fix the typos and format issues.

**Strengths And Weaknesses:**

## Strength

The paper focuses on an important problem—identifying and mitigating model failure modes.

## Weaknesses

### Weaknesses of using ChatGPT to enumerate uncommon backgrounds

**[Generalization of unseen background categories]** While ChatGPT may generate a few uncommon backgrounds (Table 1), whether the model, which mitigates the failures caused by uncommon backgrounds listed by ChatGPT, can generalize to other unseen backgrounds that are not enumerated by ChatGPT.

**[Generalization to unknown shortcut types]** The proposed method has the limitation of assuming the type of the failure mode (e.g., background). However, this leaves unknown types of failure modes (i.e., shortcuts) to be unidentified. Li et al. [1] show that only mitigating one shortcut type leads to a serious problem—amplifying other shortcuts (that are unidentified). Although the paper mentions that the method can be extended to color and size correlations (Page 3, Section 2), the limitation of relying on humans to enumerate shortcut types is unaddressed.

### Weaknesses of using CLIP to pseudo-label uncommon backgrounds for misclassified labels

**[Missclassified images with backgrounds not enumerated by ChatGPT]** The paper uses CLIP to assign uncommon backgrounds to misclassified images. However, what if the misclassified image has a background that does not belong to the background classes enumerated by ChatGPT? This is a reasonable concern because Table 1 shows that only a few background classes are enumerated by ChatGPT. As a result.

**[Problem of only using misclassified images]** Only using misclassified images has the problem of ignoring the failure modes of “correct for wrong reasons.” For example, [1] found a watermark shortcut in ImageNet’s carton class images where images with watermarks are correctly predicted using the wrong reason (i.e., watermark). Therefore, the method will fail to detect such failure modes when only using misclassified images.

**[Failure modes of CLIP]**: Using CLIP to classify background has a limitation—CLIP can also have failure modes by using other spurious cues (e.g., objects) to make predictions.

**[Split of debug seed and debug heldout]** I am confused by the construction of “debug seed” and “debug heldout” sets (Section 4.2, page 5). Can authors explain the difference between these two sets and how do you split the debug set into these two sets? For example, is it a random split? Besides, what are the purposes of these two sets?

### Weakness of mitigation method

**[Why retraining linear head?]** The proposed method retrains the linear head (Section 4.4) to mitigate model failures. I wonder what is the motivation for only retraining the linear head? Why not retrain the entire model? Besides, the paper claims to perform mitigation for the CLIP model (the 3rd line in Section 5.3.1) where the linear head does not exist. It is unclear how to apply linear head retraining for the CLIP model.

**[Failure modes of Stable Diffusion for generating synthetic training data]**  Stable Diffusion can also have failure modes of not faithfully following the prompt. The synthesized images may contain incorrect backgrounds or objects inconsistent with the input prompt.

### Weaknesses of Experiments

**[Lack of Results on Existing Benchmarks]**  The paper only conducts experiments created in this paper and fail to use many existing model failure identification and mitigation benchmarks, such as SDM benchmark in Domino (Eyuboglu et al., 2022), Salient ImageNet (Singla & Fezi, 2022), ImageNet-9 (Xiao et al. 2021), Waterbirds (Sagawa et al., 2020), UrbanCars [1], ImageNet-W [1], SpotCheck [2]). Especially, since the paper focuses on mitigating the background failure modes, the paper should particularly use two datasets where background robustness is rigorously evaluated: Salient ImageNet  (Singla & Fezi, 2022) and ImageNet-9 (Xiao et al. 2021).

**[No quantitative comparison with other failure identification or mitigation methods]** Except for the baseline (i.e., not using failure mode mitigation), the paper does not discuss or compare with other failure modes identification and mitigation methods, including Domino (Eyuboglu et al., 2022), (Jain et al., 2023), LANCE [3], bias-to-text [4], Li et al. [5], DebiAN [6], UDIS [7], George [8], Spotlight [9], StylEx [10], JTT [11], EIIL [12], LfF [13], ZOOM [14], PromptAttack [15], PGI [16], etc.

**[Results of CLIP]**  The paper mentions that the results of CLIP are shown in the appendix (the 3rd line in Section 5.3.1). However, I cannot find the results of CLIP in the appendix.

### Writing Needs Improvement

The writing of the paper has a lot of room for improvement with many typos or formatting issues.

- Figure 1: {class\\_name} -> {class_name}
- End of Section 2: Cifar10 -> CIFAR10
- End of Section 2: Cifar100 -> CIFAR100
- Citation formats in Section 3.2: Redundant author name, e.g., “Singla et al. Singla et al. (2022)”, “Kattakinda et al. Kattakinda et al. (2022)”, etc.
- Section 4.3 and Figure 1: Clip -> CLIP
- Section 4.3: wrong reference for Stable Diffusion: Ho et al. (2021) -> Rombach et al. (2022)
- Section 5.1: Imagenet -> ImageNet
- Section diffusers: missing citation to [17]
- Section 5.2: individual and Collective failure inspection -> individual and collective failure inspection.
- Section 5.2.2: however, we take a step further -> However, we take a step further
- Table 2: resnet18 -> ResNet18, ViTb16 -> ViT-B/16, …
- Many references have outdated venues or years:
  - Sabri Eyuboglu, Maya Varma, Khaled Saab, Jean-Benoit Delbrouck, Christopher Lee-Messer, Jared Dunnmon, James Zou, and Christopher Ré. Domino: Discovering systematic errors with cross-modal embeddings. arXiv preprint arXiv:2203.14960, 2022. -> ICLR 2022
  - Dan Hendrycks and Thomas Dietterich. Benchmarking neural network robustness to common corruptions and perturbations. arXiv preprint arXiv:1903.12261, 2019. -> ICLR 2019
  - Saachi Jain, Hannah Lawrence, Ankur Moitra, and Aleksander Madry. Distilling model failures as directions in latent space. arXiv preprint arXiv:2206.14754, 2022. -> ICLR 2023
  - Shiori Sagawa, Pang Wei Koh, Tatsunori B Hashimoto, and Percy Liang. Distributionally robust neural networks for group shifts: On the importance of regularization for worst-case generalization. arXiv preprint arXiv:1911.08731, 2019. -> ICLR 2020
  - Sahil Singla and Soheil Feizi. Salient imagenet: How to discover spurious features in deep learning? arXiv preprint arXiv:2110.04301, 2021. -> ICLR 2022
  - Sahil Singla, Atoosa Malemir Chegini, Mazda Moayeri, and Soheil Feiz. Data-centric debugging: mitigating model failures via targeted data collection. arXiv preprint arXiv:2211.09859, 2022. -> WACV 2024
  - Kai Xiao, Logan Engstrom, Andrew Ilyas, and Aleksander Madry. Noise or signal: The role of image backgrounds in object recognition. arXiv preprint arXiv:2006.09994, 2020. -> ICLR 2021


### References

[1] Zhiheng Li, Ivan Evtimov, Albert Gordo, Caner Hazirbas, Tal Hassner, Cristian Canton Ferrer, Chenliang Xu, and Mark Ibrahim, “A Whac-A-Mole Dilemma: Shortcuts Come in Multiples Where Mitigating One Amplifies Others,” in CVPR, 2023.

[2] Gregory Plumb, Nari Johnson, Angel Cabrera, and Ameet Talwalkar, “Towards a More Rigorous Science of Blindspot Discovery in Image Classification Models,” TMLR, 2023,

[3] Viraj Prabhu, Sriram Yenamandra, Prithvijit Chattopadhyay, and Judy Hoffman, “LANCE: Stress-testing Visual Models by Generating Language-guided Counterfactual Images,” in NeurIPS, 2023.

[4] Younghyun Kim, Sangwoo Mo, Minkyu Kim, Kyungmin Lee, Jaeho Lee, and Jinwoo Shin, “Bias-to-Text: Debiasing Unknown Visual Biases through Language Interpretation.” arXiv, 2023.

[5] Zhiheng Li and Chenliang Xu, “Discover the Unknown Biased Attribute of an Image Classifier,” in ICCV, 2021.

[6] Zhiheng Li, Anthony Hoogs, and Chenliang Xu, “Discover and Mitigate Unknown Biases with Debiasing Alternate Networks,” in ECCV, 2022.

[7] Arvindkumar Krishnakumar, Viraj Prabhu, Sruthi Sudhakar, and Judy Hoffman, “UDIS: Unsupervised Discovery of Bias in Deep Visual Recognition Models,” in BMVC, 2021.

[8] Nimit S. Sohoni, Jared A. Dunnmon, Geoffrey Angus, Albert Gu, and Christopher Ré, “No Subclass Left Behind: Fine-Grained Robustness in Coarse-Grained Classification Problems,” in NeurIPS, 2020.

[9] Greg d’Eon, Jason d’Eon, James R. Wright, and Kevin Leyton-Brown, “The Spotlight: A General Method for Discovering Systematic Errors in Deep Learning Models,” in FAccT, 2022.

[10] Oran Lang, Yossi Gandelsman, Michal Yarom, Yoav Wald, Gal Elidan, Avinatan Hassidim, William T. Freeman, Phillip Isola, et al., “Explaining in Style: Training a GAN to explain a classifier in StyleSpace,” in ICCV, 2021.

[11] Evan Z. Liu, Behzad Haghgoo, Annie S. Chen, Aditi Raghunathan, Pang Wei Koh, Shiori Sagawa, Percy Liang, and Chelsea Finn, “Just Train Twice: Improving Group Robustness without Training Group Information,” in ICML, 2021.

[12] Elliot Creager, Joern-Henrik Jacobsen, and Richard Zemel, “Environment Inference for Invariant Learning,” in ICML, 2021.

[13] Junhyun Nam, Hyuntak Cha, Sungsoo Ahn, Jaeho Lee, and Jinwoo Shin, “Learning from Failure: Training Debiased Classiﬁer from Biased Classiﬁer,” in NeurIPS, 2020.

[14] Jinqi Luo, Zhaoning Wang, Chen Henry Wu, Dong Huang, and Fernando De la Torre, “Zero-shot Model Diagnosis,” in CVPR, 2023.

[15] Jan Hendrik Metzen, Robin Hutmacher, N. Grace Hua, Valentyn Boreiko, and Dan Zhang, “Identification of Systematic Errors of Image Classifiers on Rare Subgroups,” in ICCV, 2023.

[16] Faruk Ahmed, Yoshua Bengio, Harm van Seijen, and Aaron Courville, “Systematic generalisation with group invariant predictions,” in ICLR, 2021.

[17] Patrick von Platen, Suraj Patil, Anton Lozhkov, Pedro Cuenca, Nathan Lambert, Kashif Rasul, Mishig Davaadorj, and Thomas Wolf, “Diffusers: State-of-the-art diffusion models.” GitHub, 2022.

---

> ### Author Response · Authors · 2024-02-12
>
> >Evaluate the method on existing benchmarks.
>
> Except for ImageNet results, we also show the results on CIFAR-100 (color spurious associations), and iNaturalist-2018 (Background associations). We also include the results for clean and failure accuracy separately to give some insights into our method's performance on each of this sub-categories individually for better analysis.
>
> >Discuss, cite, and compare with existing failure identification and mitigation methods.
>
> We include a section (B: Comparing to baselines) in the appendix that we compare the results with other baselines.
>
> >Add the results of CLIP as mentioned in the paper.
>
> We thank the reviewer for bringing this into our attention. We added the results for this model to our paper. It was missed by mistake. We also include them below for your convenience. However, the table in the paper is also updated with this result.
>
> | Models |  |  |  |  Accuracies |  |  |  |  |  |
> | :---: | :---: | :---: | :---: | :---: | :---: | :---: | :---: | :---: | :---: |
> | Model category | model name | Accuracy  before debugging |  | Accuracy of Individual Debugging (ours) |  |  | Accuracy of Random debugging |  |  |
> |  |  | Test | Test | seed | heldout |  | Test | seed | heldout |
> |  | ViT-B32 | $0.5388$  | $0.5582$ | $0.1794$ | **0.1635**  | | $0.5407$ | $0.0776$ | $0.0763$|
> | CLIPs |ViT-L14 | $0.7427$  | $0.7694$ | $0.2174$ | **0.2068**  | |$0.7505$ | $0.0893$ | $0.0981$ |
> |  |RN50    | $0.5928$  | $0.6129$ | $0.1980$ | **0.1833**  | |$0.6004$ | $0.0964$ | $0.0946$           |
> |  | RN101   | $0.7532$  | $0.7751$ | $0.2566$ | **0.2142**  | |$0.7570$ | $0.1084$ | $0.1115$           |
>
> >Fix the typos and format issues.
>
> We proof read the paper and we tried to fix the typos and explain parts that were unclear according to reviewers' comments. We really appreciate your detailed suggestions regarding this manner.
>
>
> [1] Saachi Jain, Hannah Lawrence, Ankur Moitra, and Aleksander Madry. Distilling model failures as directions
> in latent space. International Conference on Learning Representation (ICLR), 2023.
> [2] Sahil Singla, Atoosa Malemir Chegini, Mazda Moayeri, and Soheil Feizi. Data-centric debugging: mitigating
> model failures via targeted image retrieval. In Proceedings of the IEEE/CVF Winter Conference on
> Applications of Computer Vision, pp. 63–74, 2024.

---

> > ### Comment · Reviewer_ZC7d · 2024-02-19
> >
> > I have read all reviewers' comments and the authors' responses to other reviewers' comments. The authors’ response does not address my concerns.
> >
> > - Unknown shortcut type: The authors argue that mitigating only one shortcut (i.e., background) has benefits. However, it has a serious pitfall [1]—amplifying unknown types of shortcuts, which the authors do not acknowledge.
> > - Unknown backgrounds not listed by ChatGPT: The authors' response fails to recognize the bias discovery methods that do NOT use either (1) “a predetermined set of samples derived from a specified spurious correlation” or any external models (e.g., ChatGPT) [2,5,6,7,8,9,10,11,12,13,15,16]. Thus, the authors' argument is not convincing.
> > - Problem of only using misclassified images: I am confused by the argument of challenges of time and memory efficiency. Note that only using misclassified images fundamentally ignores the correctly predicted images for the wrong reasons, e.g., the watermark shortcut in ImageNet [1]. Thus, this is the fundamental weakness of the proposed method.
> > - Failure modes of CLIP: I appreciate the authors’ effort in verifying CLIP’s background predictions. However, 100 predictions are still not sufficient and cannot serve as the reason that CLIP won’t make wrong predictions due to other types of spurious correlations.
> > - Evaluation on existing datasets:
> >   - The ImageNet results in the paper are a customized setting (“30 data points per class” mentioned in Section 5.1) instead of the standard setting. I suggest the authors use the standard benchmarks on ImageNet, e.g., Salient ImageNet (Singla & Fezi, 2022), ImageNet-9 Background Challenge (Xiao et al. 2021), and ImageNet-W [1].
> >   - I appreciate the authors for adding the results on CIFAR-100, and iNaturalist-2018 datasets. However, I suggest the authors use standard benchmarks that most previous approaches use, e.g., the SDM benchmark in Domino (Eyuboglu et al., 2022), Waterbirds (Sagawa et al., 2020), and UrbanCars [1].
> > - Existing failure identification and mitigation methods: I appreciate the authors for adding the results in Appendix B. However, only adding the results of two methods is still insufficient. Besides, the results are still using the customized settings instead of the aforementioned public benchmarks. Finally, the revision still fails to cite and discuss the papers mentioned above [1-17].

---

### Author Response · Authors · 2024-02-12
**Revised version submitted**

We appreciate the valuable feedback provided by the reviewers, which greatly contributed to enhancing our paper. Following their comments, we have thoroughly revised the manuscript before resubmitting. We welcome any further suggestions or feedback you may have upon reviewing the updated version. Thank you.

---

### Author Response · Authors · 2024-02-13

We tried to address all the concerns that reviewers had. We thank the reviewers for their helpful comments, and since today is the last day of rebuttal, we would like to hear any remaining concerns that reviewers might have. Please let us know if we can improve the paper further.

---

### Decision · Action_Editor_qqSq · 2024-02-23

**Recommendation:** Reject

**Comment:**

Despite the authors' diligent attempts to rectify issues raised during the review process, a majority of reviewers—four out of six—remain inclined towards rejection. This consensus suggests significant reservations about the paper's current state, primarily revolving around the scalability of the proposed method, its effectiveness compared to advanced augmentation techniques, and concerns about generalizability and clarity in presentation. The reviewers have acknowledged the authors' comprehensive experiments and the potential interest of the TMLR community in the findings, yet the persistent doubts regarding the paper's technical contributions and the impact of proposed improvements indicate that further refinement is necessary.

The final recommendation does not include an option for a major revision; however, 'Resubmission of Major Revision' has been selected. This recommendation for a major revision provides an opportunity for the authors to thoroughly address the concerns raised, particularly by enhancing the method's comparative analysis, demonstrating its scalability, and improving the manuscript's clarity and presentation quality.

**Audience:**

Based on the comments from the reviewers, it looks like some people who read TMLR would find the paper interesting. Even though there are still questions about how well the method works as datasets get bigger and how it compares to newer techniques, the paper's experiments and the improvements they found could catch the attention of readers interested in how to make datasets better and improve model accuracy.

**Claims And Evidence:**

The concerns raised by the reviewers suggest that the submission might not fully substantiate its claims with clear and convincing evidence, particularly regarding the scalability and efficacy of the proposed method in generating meaningful hard backgrounds across larger datasets. There's a noted absence of rigorous comparisons with advanced augmentation techniques, such as those based on Diffusion models, which raises questions about the method's novelty and effectiveness. Additionally, the submission seems to lack a detailed analysis of its impact on testing accuracy and generalizability, which is crucial for demonstrating its practical applicability and technical contributions.

**Resubmission Of Major Revision:**

The authors may consider submitting a major revision at a later time.